# Taming Generative Diffusion Prior for Universal Blind Image Restoration

**Siwei Tu**[1,†]**, Weidong Yang**[1,†]**, Ben Fei**[2,†]
[1]Fudan University, [2]Chinese University of Hong Kong
`24110240079@m.fudan.edu.cn`, `wdyang@fudan.edu.cn`, `benfei@cuhk.edu.hk`
[†] Corresponding Authors

## Abstract

Diffusion models have been widely utilized for image restoration. However, previous blind image restoration methods still need to assume the type of degradation model while leaving the parameters to be optimized, limiting their real-world applications. Therefore, we aim to tame generative diffusion prior for universal blind image restoration dubbed **BIR-D**, which utilizes an **optimizable convolutional kernel** to simulate the degradation model and dynamically update the parameters of the kernel in the diffusion steps, enabling it to achieve blind image restoration results even in various complex situations. Besides, based on mathematical reasoning, we have provided an empirical formula for the chosen of **adaptive guidance scale**, eliminating the need for a grid search for the optimal parameter. Experimentally, Our BIR-D has demonstrated superior practicality and versatility than off-the-shelf unsupervised methods across various tasks both on real-world and synthetic datasets, qualitatively and quantitatively. BIR-D is able to fulfill multi-guidance blind image restoration. Moreover, BIR-D can also restore images that undergo multiple and complicated degradations, demonstrating the practical applications. The code is available at https://github.com/Tusiwei/BIR-D

# 1  Introduction

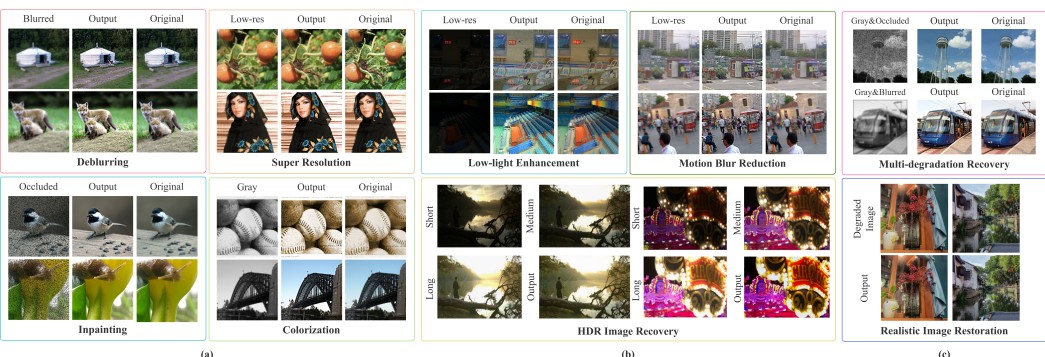

Figure 1: Blind Image Restoration Diffusion Model (BIR-D) can achieve high-quality restoration for different types of degraded images. BIR-D not only has the capability to restore (a) linear inverse problems when the degradation function is known. BIR-D can also achieve high-quality image restoration in (b) blind issues with unknown degradation functions, as well as in (c) mixed degradation and real degradation scenarios.

38th Conference on Neural Information Processing Systems (NeurIPS 2024).

Images inevitably suffer from a degradation in quality in the process of capturing, storing, and compressing. Thus, the image restoration task intends to establish a mapping between the degraded image and the original image, to recover a high-quality image from the degraded image. In an ideal scenario, the ultimate goal is to undo and restore the degradation process of the image. However, in reality, the complexity of the degradation mode often leads to the incapability to fully restore the original high-quality image, which also makes traditional supervised approaches unsuitable for all types of image restoration tasks. According to the degradation mode, image restoration tasks can be divided into two types: **non-blind** and **blind** problems. **Blind** problems, such as low light enhancement, motion blur reduction and HDR image restoration, refer to image restoration problems where the degradation functions and parameters are totally unknown.

The blind image restoration problem has attracted increasing attention with the development of generative models. The unsupervised blind image restoration methods represented by Generative Adversarial Networks (GANs) [1; 2; 3; 4] have the capability to train networks on large datasets of clean images and learn real-world knowledge. However, GANs are still difficult to avoid falling into limitations such as poor diversity and difficulty in model training. In parallel, diffusion model [5; 6; 7; 8; 9] have shown strong performance in terms of quality and diversity compared to GANs. Pioneer works such as GDP [10], DDRM [11], and DDNM [12] attempt to solve such problems by incorporating the degraded image $y$ as guidance in the sampling process of diffusion models. By modeling posterior distributions in an unsupervised sampling manner, these approaches showcase the potential for practical guidance in blind image restoration, offering promising implications for real-world applications. However, the degradation types in these models still need to be assumed, limiting the practicality of natural image restoration where the complicated degradation models always remain unknown.

To this end, we propose an effective and versatile Blind Image Restoration Diffusion Model (BIR-D). It utilizes well-trained DDPM [13] as an effective prior and is guided by degraded images to form a universal method for various image restoration and enhancement tasks. To uniformly model the unknown degradation function of blind image restoration, an optimizable convolutional kernel is dynamically optimized and utilized to simulate the degradation function at each denoising step. Specifically, BIR-D updates the convolution kernel parameters based on the gradient of distance loss between the generated image undergoing our optimizable convolutional kernel and the given degraded image. At the same time, all existing image restoration methods [10; 11; 12] that use diffusion models manually set the guidance scale as a hyperparameter to control the magnitude of guided generation, which also remains unchanged throughout the sampling process. However, for images from different tasks, the guidance scale required for each diffusion step is not entirely the same. To deal with this issue, we have derived an empirical formula for the guidance scale, which can calculate the optimal guidance scale for the next denoising step in real-time during the sampling process. This improvement avoids the need to manually grid search the optimal value of the guidance scale when solving different tasks and also enhances the quality of generated images. With the help of a well-trained DDPM, the above designs enable BIR-D to tackle various blind image restoration tasks. BIR-D can also achieve multi-degradation or multi-guidance image restoration. Furthermore, it showcases satisfactory results in addressing restoration issues related to complex degradation types encountered in real-world scenarios.

## 2   BIR-D: Universal Blind Image Restoration Diffusion Model

In this study, we aim to use a well-trained DDPM [13] to learn the prior distribution of images and ultimately solve non-blind and blind problems in various image restoration tasks.

### 2.1   Optimizable convolutional kernel as a universal degradation function

For a natural image $x$, its corresponding degraded image $y$ can be obtained by the degradation function $y = \mathcal{D}(x)$. Most of the blind image restoration methods [10; 12] are used to solve the situation where the degradation function $\mathcal{D}$ is known while leaving the parameters of $\mathcal{D}$ are unknown. However, when dealing with real-world image restoration problems, the degradation function $\mathcal{D}$ is not only an unknown quantity but also difficult to accurately represent mathematically. Therefore, we propose an optimized convolutional kernel to simulate complex degradation functions. The

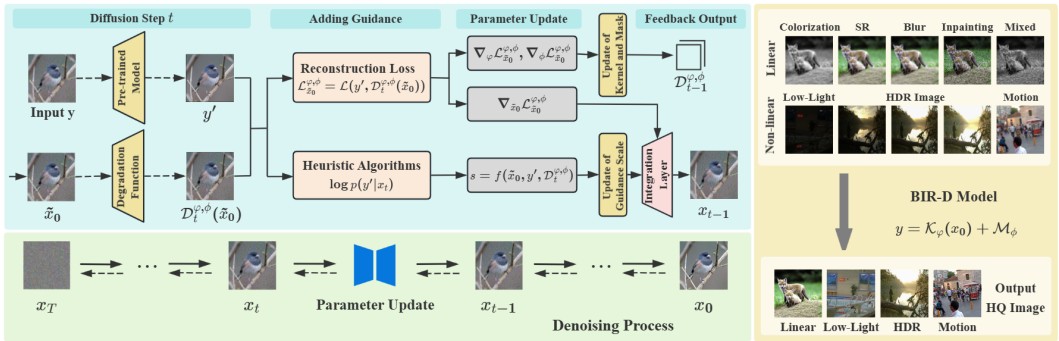

Figure 2: **Overview of BIR-D.** Degraded image $y$ was given during the sampling process. BIR-D systematically incorporates guidance from degraded images in the reverse process of the diffusion model and optimizes the degraded model at the same time. For degraded image $y$, pre-training is first performed to provide a better initial state for BIR-D. BIR-D introduces a distance function in each step of the reverse process of the diffusion model to describe the distance loss between the degraded image $y$ and the generated image $\tilde{x}_0$ after the degradation function, so that the gradient could be used to update and simulate a better degradation function. Based on the empirical formula, the adaptive guidance scale can be calculated to provide optimal guidance during the sampling process.

parameters of the convolution kernel in the degradation function are dynamically optimized along with the denoising steps.

Moreover, in the real-world scenario, considering that there are different noises in different subtle areas of the image, using only one optimized convolutional kernel may not fully cover this situation. Therefore, we propose to utilize a mask $\mathcal{M}$ to model and estimate these noises. Thus, the entire degradation process can be represented as: $y = K(x) + \mathcal{M}$, where $\mathcal{K}$ refers to the optimized convolutional kernel used in the model and $\mathcal{M}$ is a mask with the same dimension as image $x$. $\mathcal{K}$ and $\mathcal{M}$ have their own optimizable parameters, forming the degradation function $mathcalD$. In this way, any degradation process can be simulated by this degradation function.

## 2.2 Empirical formula of guidance scale

In the reverse denoising process of DDPM, the generated images can be conditioned on degraded image $y$ [39]. Specifically, the distribution $p_\theta(x_{t-1}|x_t)$ of reverse denoising is converted into

---

**Algorithm 1:** Unconditional diffusion model with the guidance of degraded image $y$, given a diffusion model noise prediction function $\epsilon_\theta(x_t, t)$.

**Input:** Degraded image $y$, degradation function $\mathcal{D}$ composed of optimized convolutional kernels $\mathcal{K}$ with parameters $\varphi$ and mask $\mathcal{M}$ with parameters $\phi$, learning rate $l$, distant measure $\mathcal{L}$.
**Output:** Output image $x_0$ conditioned on $y$.
Sample $x_T$ from $\mathcal{N}(0, I)$
**for** t from T to 1 **do**

$\tilde{x}_0 = \frac{x_t}{\sqrt{\bar{\alpha}_t}} - \frac{\sqrt{1-\bar{\alpha}_t}\epsilon_\theta(x_t,t)}{\sqrt{\bar{\alpha}_t}}$

$\mathcal{L}_{\varphi,\phi,\tilde{x}_0} = \mathcal{L}(y, \mathcal{D}^{\varphi,\phi}(\tilde{x}_0))$

$s = -\frac{(x_t-\mu)^T g + C + \log N}{\mathcal{L}(\mathcal{D}^{\varphi,\phi}(\tilde{x_0}),y)}$

$\tilde{x}_0 \leftarrow \tilde{x}_0 - \frac{s(1-\bar{\alpha}_t)}{\sqrt{\bar{\alpha}_{t-1}}\beta_t}\nabla_{\tilde{x}_0}\mathcal{L}_{\varphi,\phi,\tilde{x}_0}$

$\tilde{\mu}_t = \frac{\sqrt{\bar{\alpha}_{t-1}}\beta_t}{1-\bar{\alpha}_t}\tilde{x}_0 + \frac{\sqrt{\bar{\alpha}_t}(1-\bar{\alpha}_{t-1})}{1-\bar{\alpha}_t}x_t$

$\tilde{\beta}_t = \frac{1-\bar{\alpha}_{t-1}}{1-\bar{\alpha}_t}\beta_t$

Sample $x_{t-1}$ from $\mathcal{N}(\tilde{\mu}_t, \tilde{\beta}_t I)$

$\varphi \leftarrow \varphi - l\nabla_\varphi\mathcal{L}_{\varphi,\phi,\tilde{x}_0}$

$\phi \leftarrow \phi - l\nabla_\phi\mathcal{L}_{\varphi,\phi,\tilde{x}_0}$

**return** $x_0$

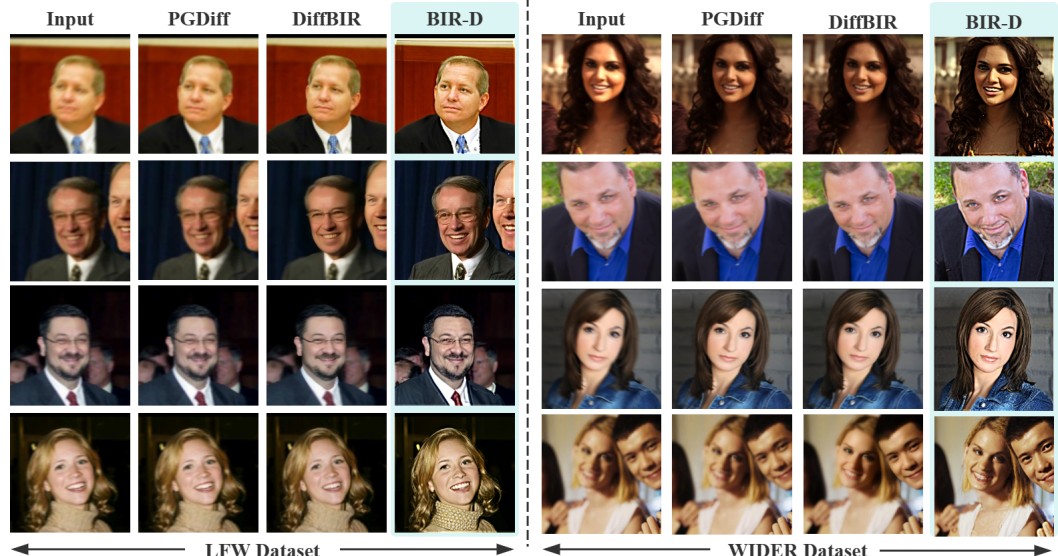

Figure 3: Comparison of image quality for blind face restoration results on LFW [14] and WIDER dataset [15].

| Task | LFW dataset | | WIDER dataset | |
|------|-------------|------|---------------|------|
| | FID | NIQE | FID | NIQE |
| PGDiff [16] | 71.62 | 4.15 | 39.17 | 3.93 |
| DiffBIR [17] | **39.58** | 4.03 | 32.35 | 3.78 |
| BIR-D | 40.12 | **3.94** | **31.49** | **3.65** |

Table 1: **Quantitative comparison of blind face restoration on LFW and WIDER datasets**

a conditional distribution $p_\theta(x_{t-1}|x_t, y)$. It is demonstrated [13] that the difference between it and the original formula lies in the addition distribution of $p(y|x_t)$, which serves as a probability representation for denoising $x_t$ into a high-quality image consistent with $y$. Previous work [10] proposed a feasible calculation to approximate this indicator by using heuristic algorithms:

$$\log p(y \mid x_t) = -\log N - s\mathcal{L}(\mathcal{D}(\tilde{x}_0), y)), \tag{1}$$

where $N$ is the normalization factor, which is the distribution $p_\theta(y|x_{t+1})$, and $s$ is the scalar factor used to control the importance of guidance, named guidance scale. $\mathcal{L}$ is the distance metric. The value of the guidance scale plays a crucial role in the quality of the image generation result. A larger value can lead to overall blurring of the image, while a smaller value can result in missing details in the restoration. However, the guidance scale in existing works [10; 9; 12] can only be manually set as a hyperparameter. But in specific experiments, the optimal value of the guidance scale varies in different masks, degraded images, and diffusion steps. The original configuration necessitates thorough testing for the initial setup. Additionally, employing the same guidance scale for every denoising step is not an optimal choice.

Therefore, we propose an empirical formula for the guidance scale, which can dynamically calculate and update the optimal values of guidance factors in real-time at each diffusion step of degraded images in specific repair tasks. Specifically, we noticed that the distribution $\log p_\theta(y|x_t)$ can be applied to perform Taylor expansion around $x = \mu$ and take the first two terms. The detailed process of proving can be found in Appendix D.

$$\log p_\theta(y \mid x_t) = (x_t - \mu)^T g + C, \tag{2}$$

where $g = \nabla_{x_t} \log p_\theta(y \mid x_t) \mid_{x_t=\mu}$, $C = \log p(y \mid x_t) \mid_{x_t=\mu}$. By combining the heuristic approximation formula and Taylor expansion formula mentioned above, we can simplify the empirical

| Task | 4×Super resolution | | | | Deblur | | | | 25% Inpainting | | | | Colorization | | | |
|---|---|---|---|---|---|---|---|---|---|---|---|---|---|---|---|---|
| | PSNR | SSIM | Consistency | FID | PSNR | SSIM | Consistency | FID | PSNR | SSIM | Consistency | FID | PSNR | SSIM | Consistency | FID |
| RED[19] | 24.18 | 0.71 | 27.57 | 98.30 | 21.30 | 0.58 | 63.20 | 69.55 | - | - | - | - | - | - | - | - |
| DGP[18] | 21.65 | 0.56 | 158.74 | 152.85 | 26.00 | 0.54 | 475.10 | 136.53 | 27.59 | 0.82 | 414.60 | 60.65 | 18.42 | 0.71 | 305.59 | 94.59 |
| SNIPS[20] | 22.38 | 0.66 | 21.38 | 154.43 | 24.73 | 0.69 | 60.11 | 17.11 | 17.55 | 0.74 | 587.90 | 103.50 | - | - | - | - |
| DDRM[11] | **26.53** | 0.78 | 19.39 | 40.75 | **35.64** | **0.98** | 50.24 | 4.78 | 34.28 | 0.95 | **4.08** | 24.09 | **22.12** | 0.91 | 37.33 | 47.05 |
| DDNM[21] | 25.36 | **0.81** | 7.52 | 39.14 | 24.66 | 0.71 | 41.70 | 4.64 | 32.16 | **0.96** | 5.42 | 17.63 | 21.95 | 0.89 | 36.41 | 38.79 |
| GDP[10] | 24.42 | 0.68 | 6.49 | 38.24 | 25.98 | 0.75 | 41.27 | 2.44 | **34.40** | **0.96** | 5.29 | 16.58 | 21.41 | **0.92** | 36.92 | 37.60 |
| BIR-D | 24.58 | 0.71 | **6.32** | **37.54** | 26.31 | 0.73 | **38.42** | **2.32** | 33.59 | 0.90 | 5.18 | **15.73** | 22.09 | 0.89 | **36.12** | **36.58** |

Table 2: **Quantitative comparison of linear inverse problems on ImageNet 1k[18].**

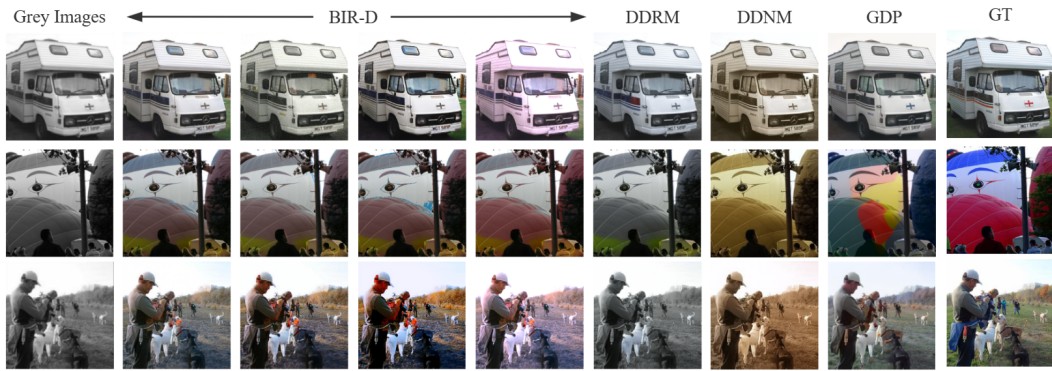

Figure 4: **Comparison of colorization image on ImageNet 1k[18].** BIR-D can generate various outputs on the same input image.

formula for the guidance scale:

$$s = -\frac{(x_t - \mu)^T g + C + \log N}{\mathcal{L}(\mathcal{D}(\tilde{x}_0), y)}. \tag{3}$$

The guidance scale is related to the generated images $x$, degraded image $y$, and the degradation function $\mathcal{D}$. This value of this **Adaptive Guidance Scale** can be dynamically updated in each diffusion step so that each step in the diffusion model can use the most appropriate guidance scale.

## 2.3 Sampling process of BIR-D

Through empirical formulas, we can obtain the conditional transition formula in the reverse process of the diffusion model.

$$\log p_\theta(x_t|x_{t+1}, y) = \log\left(p_\theta(x_t|x_{t+1})p(y|x_t)\right) + N_1 \tag{4}$$
$$\approx \log p(z) + N_2, \tag{5}$$

where $z$ conforms to the distribution $\mathcal{N}(z; \mu_\theta(x_t, t) + \Sigma g, \Sigma)$. The intermediate quantity $g = \nabla_{x_t} \log p(y|x_t)$. The value of $g$ can be obtained by calculating the gradient in heuristic algorithms in eq. (1), which includes the parameter of guidance scale:

$$g = \nabla_{x_t} \log p(y|x_t) = -s\nabla_{x_t}\mathcal{L}(\mathcal{D}(x_t), y) \tag{6}$$

The other terms $N_1$, $N_2$, and the variance of the reverse process $\Sigma = \Sigma_\theta(x_t)$ in eq. (4) and eq. (5) are constants, and the unconditional distribution $p_\theta(x_{t-1}|x_t)$ is given by traditional diffusion models.

Therefore, the conditional transition distribution $p(x_{t-1}|x_t, y)$ can be approximately estimated by adding $-(s\Sigma\nabla_{x_t}\mathcal{L}(\mathcal{D}(x_t), y))$ to the mean of the traditional unconditional transition distribution. Previous studies [10] have shown that the addition of $\Sigma$ has a negative impact on the quality of generated images. Therefore, in this experiment, we omitted the term $\Sigma$, and the complete sampling process is shown in algorithm 1.

Detailly, in the diffusion step $t$ of the sampling process, the noise of $x_t$ is first predicted from the given pre-trained DDPM and eliminated to obtain an estimated value of $x_0$. Subsequently, apply the degradation function of step $t$ to $x_0$ and calculate its reconstruction loss with the degraded image

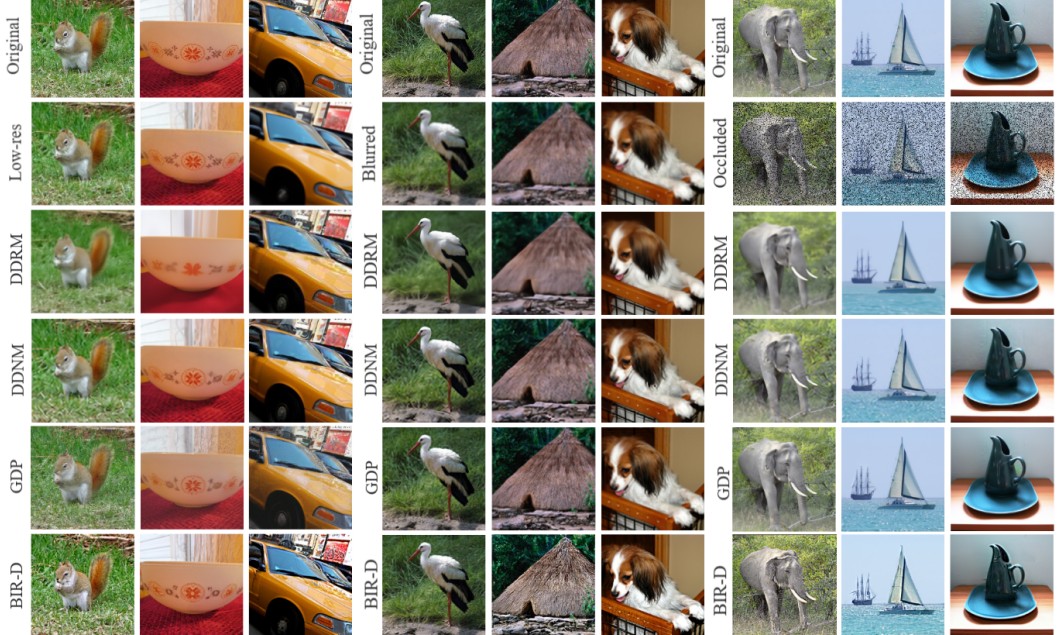

**(a) 4 × Super Resolution**   **(b) Deblurring**   **(c) 25% Inpainting**

Figure 5: Results of linear degradation tasks on 256 × 256 images from ImageNet 1k.

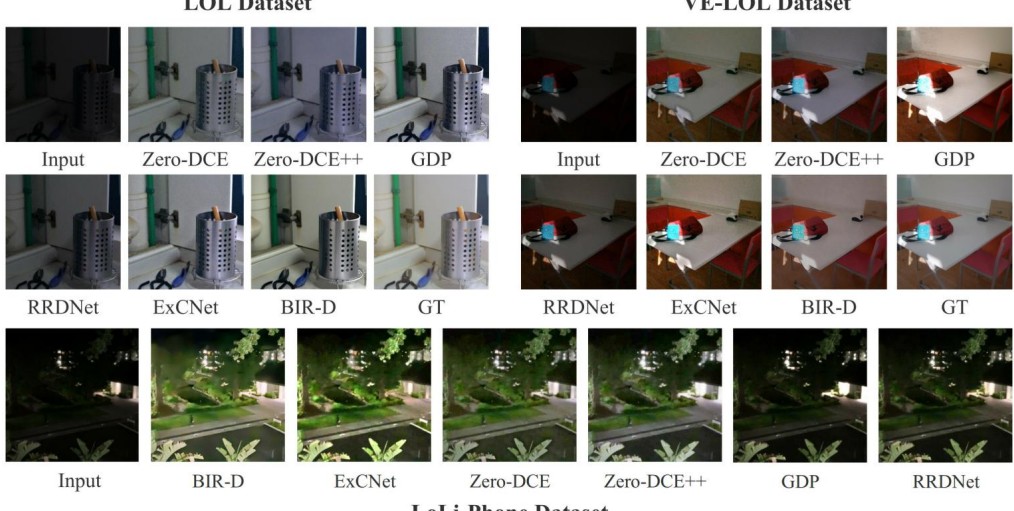

Figure 6: Comparison of image quality in low-light enhancement task on the LoL [22], VE-LOL [23] and LoLi-Phone [24] datasets.

$y$. We utilize our adaptive guidance scale for sampling the next step latent $x_{t-1}$. In this process, it is necessary to calculate the gradient about $x_0$ and the parameters of each convolution kernel in the distance metric loss, which is used to update the convolution kernel parameters in real time for the next sampling process.

**Pre-process.** The empirical formula for the guidance scale we construct is related to the degradation function. Herein, when the model simulates the degradation function more reasonably, BIR-D can obtain more appropriate guidance scale values accordingly. To this end, we introduce a first-stage pre-training model from [17] to further enhance the model's capability to correct initial deviations. This enables the model to have a strong correction ability for significant deviations in the degradation function during the initial diffusion step, ultimately generating ideal image restoration results.

| Task | LOL | | | | | VE-LOL-L | | | | |
|------|-----|-----|-----|-----|-----|---------|-----|-----|-----|-----|
| | PSNR | SSIM | LOE | FID | PI | PSNR | SSIM | LOE | FID | PI |
| ExCNet[25] | **16.04** | 0.62 | 220.38 | 111.18 | 8.70 | 16.20 | 0.66 | 225.15 | 115.24 | 8.62 |
| Zero-DCE[26] | 14.91 | **0.70** | 245.54 | 81.11 | 8.84 | **17.84** | **0.73** | 194.10 | 85.72 | 8.12 |
| Zero-DCE++[27] | 14.86 | 0.62 | 302.06 | 86.22 | 7.08 | 16.12 | 0.45 | 313.50 | 86.96 | 7.92 |
| RRDNet[28] | 11.37 | 0.53 | 127.22 | 89.09 | 8.17 | 13.99 | 0.58 | 94.23 | 83.41 | 7.36 |
| GDP[10] | 13.93 | 0.63 | 110.39 | 75.16 | 6.47 | 13.04 | 0.55 | 79.08 | 78.74 | 6.47 |
| BIR-D | 14.52 | 0.56 | **105.42** | **68.98** | **4.87** | 13.87 | 0.51 | **78.18** | **74.54** | **5.73** |

Table 3: **Quantitative comparison among various zero-shot learning methods of low-light enhancement task on LOL [22] and VE-LOL-L [23]** Bold font represents the best metric result.

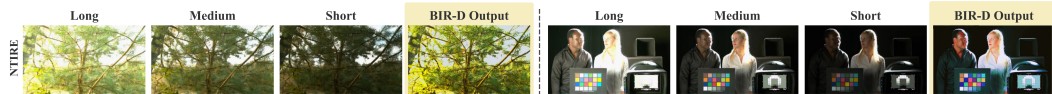

Figure 7: Comparison of image quality for HDR image recovery results on NTIRE [29].

**Multi-degradation Image Restoration.** In the real world, the degradation process often involves a combination of multiple different complex types. To improve the image restoration capability of the model in complex situations and enhance its practicality, we propose to extend BIR-D into multi-task scenarios. To our surprise, BIR-D can fulfill multi-degradation image restoration without any modification (Figure 9) thanks to the mixture of degradation types can also be simulated as an unknown degradation by an optimizable convolutional kernel.

## 3 Experiments

In this section, we systematically compare BIR-D with other blind image restoration methods in real-world and synthetic datasets. We have attached some more specific details, such as the dataset, implementation, evaluation, and other results in the Appendix.

**Blind Image Restoration on Real-world Datasets.** Firstly, we evaluate the blind image restoration capability of BIR-D on two real-world datasets, namely LFW dataset [14] and WIDER dataset [15]. As shown in Figure 3, BIR-D successfully simulated and removed blur, and achieved more ideal facial detail restoration. The quantitative results in Table 1 shows that BIR-D outperforms PGDiff [16] and DiffBIR [17] in NIQE metric on both datasets and FID metric on WIDER, demonstrating better blind image restoration performance.

**Comparison on Common Linear Inverse Problems.** We conducted experiments on linear inverse problems on ImageNet 1k to compare BIR-D with off-the-shelf methods. For each experiment, we calculated the average Peak Signal-to-Noise Ratio (PSNR), Structural Similarity (SSIM), Consistency, and FID results, where PSNR, SSIM, and Consistency are used to quantify the faithfulness between the generated image and the original image, while FID is used to measure the quality of the generated image. To make fair comparisons, other methods are given known degradation functions as reported in the original paper while BIR-D utilizes universal degradation functions for different tasks. Table 2 shows that BIR-D outperforms other methods in terms of Consistency and FID in almost all tasks. As shown in Figure 5, the images generated by BIR-D demonstrate a high level of image quality and details. Moreover, Figure 4 also demonstrates that BIR-D can generate various results in image restoration tasks.

**Low Light Enhancement.** We further evaluated the effectiveness of BIR-D in low-light image enhancement. Following the previous works [10], we utilized three datasets, LOL [22], VE-LOL-L [23], and LoLi-Phone [24], to test the restoration ability of BIR-D. As shown in Table 3, our BIR-D outperforms all the zero-shot methods in both FID and Lightness Order Error (LOE) [40], and demonstrates significant improvement in Perceptual Index (PI) [41]. A lower PI value reflects better perceptual quality, while a lower LOE reflects a better natural preservation ability of the generated image, making images to have a more natural sensory experience. As shown in Figure 6 and the Appendix, BIR-D exhibits reasonable and well-exposed results.

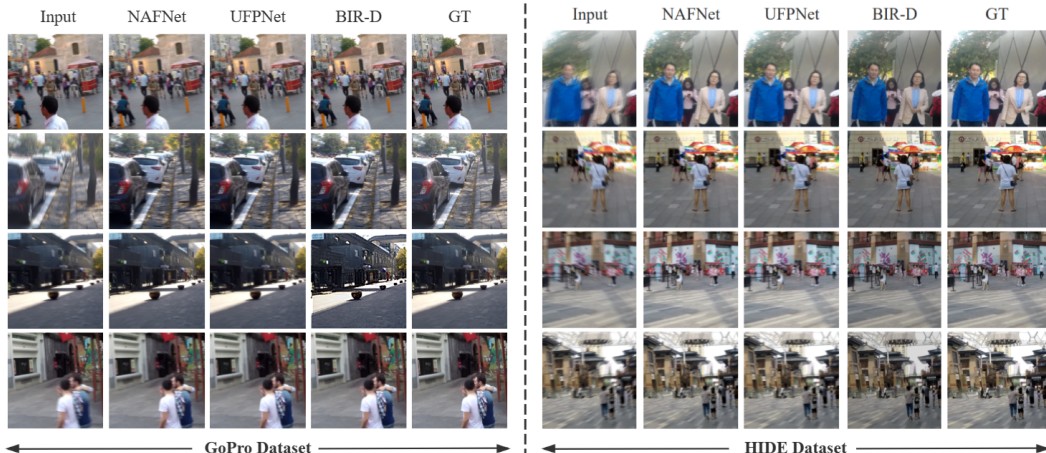

Figure 8: Comparison of image quality for motion blur reduction results on GoPro [30] and HIDE dataset [31].

| Motion Blur Reduction | GoPro | | HIDE | | HDR Recovery | NTIRE | | | |
|---|---|---|---|---|---|---|---|---|---|
| | PSNR | SSIM | PSNR | SSIM | | PSNR | SSIM | LPIPS | FID |
| DeepRFT[32] | 33.23 | 0.963 | 31.42 | 0.944 | Deep-HDR[33] | 21.66 | 0.76 | 0.26 | 57.52 |
| MSDI-Net[34] | 33.28 | 0.964 | 31.02 | 0.940 | AHDRNet[35] | 18.72 | 0.58 | 0.39 | 81.98 |
| NAFNet[36] | 33.69 | 0.967 | 31.32 | 0.943 | HDR-GAN[37] | 21.67 | 0.74 | 0.26 | 52.71 |
| UFPNet[38] | 34.06 | **0.968** | 31.74 | 0.947 | GDP[10] | 24.88 | 0.86 | **0.13** | 50.05 |
| BIR-D | **34.12** | **0.968** | **32.09** | **0.948** | BIR-D | **25.03** | **0.88** | 0.16 | **48.74** |

Table 4: **Quantitative comparison of motion blur reduction and HDR image recovery tasks.**

**HDR Image Recovery.** In the HDR image restoration task, we compared BIR-D with other leading methods, including DeepHDR [33], AHDRNet [35], HDR-GAN [37], and GDP [10], on the NTIRE2021 Multi-Frame HDR Challenge [29] dataset. The quantitative and qualitative results are presented in Table 4 and Figure 7, with BIR-D showing the best PSNR and SSIM levels, and successfully generating results with rich and accurate detailed information.

**Motion Blur Reduction.** To evaluate the performance of BIR-D in the motion blur reduction tasks, we compare BIR-D with the state-of-the-art motion blur reduction methods on GoPro dataset [30] and HIDE dataset [31]. We used the same input image, which also means that the motion blur of the input image is the same, ensuring fairness in comparison. The comparison results of the metrics are presented in Table 4, where BIR-D outperforms existing methods in both PSNR and SSIM. As shown in Figure 8, BIR-D can effectively achieve the elimination of motion blur. The generated images not only achieve a better quality but also receive restoration with more clear details.

**Multi-Degradation Image Restoration.** Encouraged by the excellent restoration performance of BIR-D on single restoration task, we further tested the image restoration performance of BIR-D in solving multi-task image restoration. As shown in Figure 9, we take a degraded image on the ImageNet

| Methods | Dynamic Update | | LOL | | | | | LoLi-Phone | |
|---|---|---|---|---|---|---|---|---|---|
| | Kernel | Guidance Scale | PSNR | SSIM | LOE | FID | PI | LOE | PI |
| Model A | ✗ | ✗ | 8.96 | 0.46 | 210.88 | 113.36 | 8.24 | 110.05 | 8.36 |
| Model B | ✗ | ✓ | 9.58 | 0.48 | 203.83 | 102.47 | 7.90 | 102.55 | 8.25 |
| Model C | ✓ | ✗ | 14.35 | 0.54 | 113.56 | 82.14 | 5.23 | 75.34 | 7.94 |
| BIR-D | ✓ | ✓ | **14.52** | **0.56** | **105.42** | **68.98** | **4.87** | **72.83** | **6.12** |

Table 5: **The ablation study on the optimizable convolutional kernel and the empirical settings of guidance scale.**

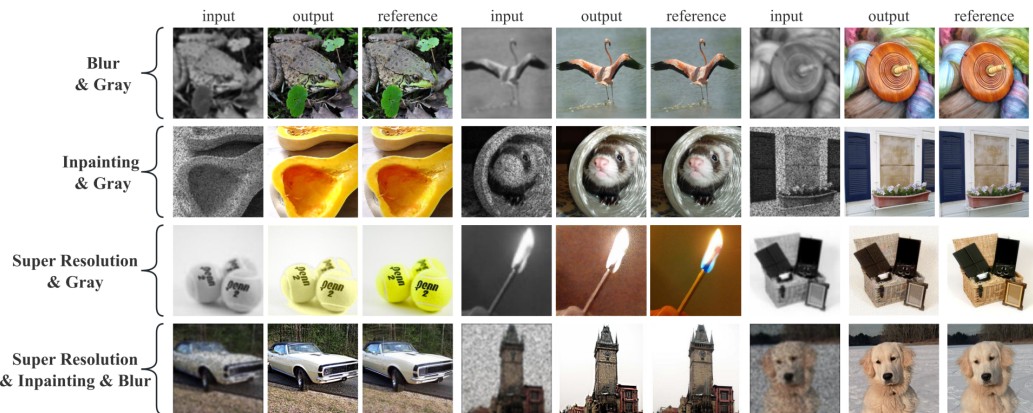

Figure 9: Results of multi-task image restoration.

| Task | Random initial value | | | | Biased initial value | | | |
|---|---|---|---|---|---|---|---|---|
| | PSNR | SSIM | Consistency | FID | PSNR | SSIM | Consistency | FID |
| BIR-D without pre-training model | 25.88 | 0.69 | 40.24 | 2.55 | 21.49 | 0.61 | 53.78 | 4.32 |
| BIR-D | 26.31 | 0.73 | 38.42 | 2.32 | 25.97 | 0.71 | 39.87 | 2.41 |

Table 6: **The ablation study on the effectiveness of the pre-training model.**

dataset where two types of degradation are mixed as an example. The optimizable convolution kernel of BIR-D can also simulate these complicated degradation functions. The generated images obtained have excellent results in both image quality and details.

## 4  Ablation study

**The Effectiveness of Optimizable Convolutional Kernel and Adaptive Guidance Scale.** The ablation studies on the real-time optimizable convolutional kernel parameters and guidance scale were performed to reveal the effectiveness of these settings. We further tested the LOL [22] and the most challenging LoLi-phone [24] datasets. Model A fixed the convolutional kernel parameters and guidance scale. Models B and C represent fixed parameters for the convolution kernel and the fixed guidance scale, respectively. As illustrated in Figure 10, the fixed guidance scale with a bias set at $s = 80000$ resulted in the emergence of mineral textures in the images. By contrast, as shown in Table 5, BIR-D outperformed other models in all indicators, demonstrating the effectiveness of an optimizable convolutional kernel and adaptive guidance scale.

**The Effectiveness of the First Stage Pre-training Model.** We conducted further experiments on the deblur task to demonstrate the impact of the first-state pre-training model. As shown in Table 6, for a randomly initialized convolution kernel parameter, all metrics of BIR-D were better than BIR-D without the pre-training model. These results indicate that the first-stage pre-trained model is able to provide better initial state of images for our BIR-D.

## 5  Parameter analysis

**The Parameter Variations of the Optimizable Convolution Kernel and Mask in the Reverse Steps.** In order to visualize the variation trends of the parameters of convolution kernel mask in the reverse process, we conducted experiments on the test set of the LOL dataset from the low-light enhancement task. As shown in Figure 11(a), the mean values of the convolution kernel parameters and degradation mask are given by random initialization and gradually increase with the progress of the time steps. This increase in magnitude is influenced by the gradient of the distance metric with respect to the corresponding parameters. When the sampling step $t<500$, the difference between $\tilde{x}_0$ and $y$ changes slightly, resulting in correspondingly smaller gradient values.

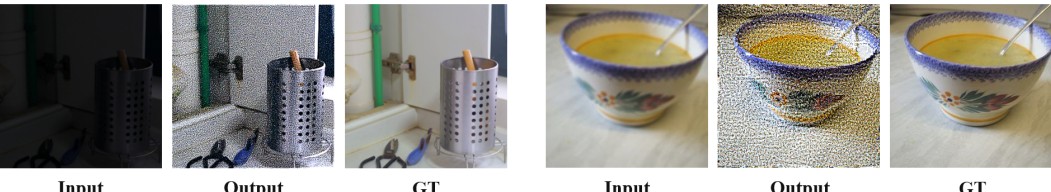

| Input | Output | GT | Input | Output | GT |

Figure 10: Qualitative results when the fixed guidance scale is biased towards a larger value of $s = 80000$.

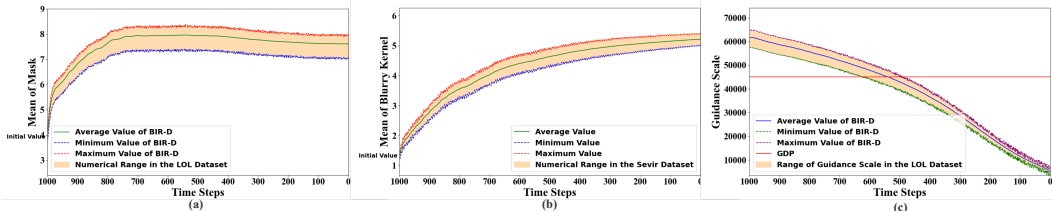

Figure 11: Illustration of **(a)** the variation of the mean of optimizable convolutional kernel parameters in each step of the sampling process. **(b)** The variation of the mean of degradation mask in each step of the sampling process. **(c)** The variation of adaptive guidance scale in each step of the sampling process. These experiments are performed on LOL dataset.

BIR-D employs masks in the degradation function with the intent to address the image restoration of local regions characterized by substantial shifts in brightness. Figure 11(b) shows that the mask $\mathcal{M}$ of the degradation model has an upward trend from their initial values, making the overall degradation function approach the true degradation. As shown in Figure 12, during the sampling process, the degradation mask learns the detailed information of the image, including local regions with significant brightness differences. This process is obtained by updating the gradient of the distance metric with respect to the degradation mask parameters.

**The Theoretical Analysis of the Changing Trend of Guidance Scale in the Reverse Steps.** We take the variation in the guidance scale of BIR-D on the LOL dataset as an example to analyze the trend of its changes during the reverse steps. As shown in Figure 11(c), the guidance scale gradually decreases with the sampling step, which aligns with the actual situation. When the sampling step $t<500$, as $t$ decreases, the difference between $x_t$ and $x_{t-1}$ decreases with decreasing $t$, indicating a reduction in the simulated noise at each step. Therefore, the level of guidance required for each sampling step should also be reduced accordingly, leading to a decrease in the required guidance scale values. According to Equation (3), when step $t$ is small, the gradient term $g$ also decreases due to the small change in $x_t$ at each step. The speed of the gradient term decreases is greater than the speed of distance metric decreases, resulting in a decrease in the value of the guidance scale.

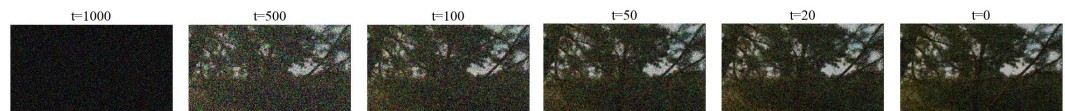

Figure 12: The changing of degradation mask during the sampling process in HDR recovery.

# 6 Conclusion

In this paper, we propose Blind Image Restoration Diffusion, which is a unified model that can be used to solve various blind image restoration problems. We utilize optimized convolutional kernels to simulate and update the degradation function in the diffusion step in real time, and derive the empirical formula of the guidance scale in detail, so that it can better utilize the unconditional diffusion model to generate high-quality images. The ability to solve various blind image restoration tasks, including low-light enhancement and motion blur reduction, has also been verified through various indicators of datasets.

## Acknowledgment

The authors would like to thank Zhaoyang Lyu for his technical assistance. This work was supported by the National Natural Science Foundation of China (U2033209)

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

In this appendix, we provide a more detailed derivation process for adaptive guidance scale and more image restoration results. Appendix A provides more blind image restoration results for images undergoing various multi-degradation modes. Appendix B is a preliminary for the diffusion model. Appendix C introduces the related work, including an introduction to diffusion-based image restoration models and existing leading methods in the field of blind image restoration. Appendix D is the complete derivation process of the empirical setting of the adaptive guidance scale. Appendix E proposes the algorithmic process of BIR-D for accomplishing multi-guidance blind image restoration. Appendix F conducts further ablation experiments on the size and other parameters of the convolution kernel and provides the optimal convolution kernel parameter settings for different image restoration tasks. Appendix G presents and analyzes the variations of the mask parameters in the reverse steps of LOL dataset. Appendix H presents the results of blind image restoration tasks, including blind face restoration, low-light enhancement, motion blur reduction, and HDR image restoration. Appendix I provides more results on the ImageNet dataset for four basic tasks, including super-resolution, colorization, deblurring and inpainting, as well as corresponding experimental details and parameter settings. Appendix J discusses limitations and future works for BIR-D.

## A  Multi-degradation Image Restoration

In this section, we attempted more complex degradation scenarios to test the image restoration ability of BIR-D. The input images are derived from adding different degradation types, which include $4 \times$ super-resolution, colorization, deblurring, and inpainting. As shown in Figure 13, BIR-D achieved satisfactory restoration results in four different multi-task image restoration tasks, which indicates the image restoration capability of BIR-D under complex degraded scenarios.

## B  Preliminary

Diffusion models consists of the forward and reverse processes. The forward process continuously adds noise to a natural image $x_0$ through $T$ diffusion steps to obtain the noise distribution $x_T \sim \mathcal{N}(0, I)$, where $\mathcal{N}$ represents the Gaussian distribution. The reverse process aims to simulate the noise in each diffusion step and eliminate it, ultimately obtaining the restored generated image $x_0$.

The forward process is a Markov chain defined by the following equation:

$$q(x_1, \cdots, x_T | x_0) = \prod_{t=1}^{T} q(x_t | x_{t-1}) \tag{7}$$

It corrupts the initial data $x_0$ into distribution $x_T$ that is close to Gaussian noise after $T$ steps of diffusion, with each sample process defined by $q(x_t | x_{t-1}) = \mathcal{N}(x_t; \sqrt{1 - \beta_t} x_{t-1}, \beta_t I)$, where $\beta_t$ is the variance of a forward process. The variance can be set as a constant or learned by reparameterization. Simultaneously defining $\alpha_t = 1 - \beta_t, \bar{\alpha}_t = \prod_{i=1}^{t} \alpha_i$. It has been proven by [42] that through mathematical reasoning, $x_t$ at any diffusion step can be directly calculated from the starting $x_0$:

$$x_t = \sqrt{\bar{\alpha}_t} x_0 + \sqrt{1 - \bar{\alpha}_t} \epsilon, \tag{8}$$

where $\epsilon \sim \mathcal{N}(0, I)$. When $T$ is large enough, $\sqrt{\bar{\alpha}_t}$ approaches 0, and at this point $q(x_T | x_0)$ is closer to the latent distribution of $x_T$.

The reverse process is also a Markov chain, which gradually denoises a standard multivariate Gaussian distribution into a denoised image $x_0$. Firstly, sample $x_t \sim \mathcal{N}(0, I)$. The conditional distribution of the reverse process is $p_\theta(x_{t-1} | x_t) = \mathcal{N}(x_{t-1}; \mu_\theta(x_t, t), \Sigma_\theta I)$. According to the Bayesian formula, it can be transformed as follows:

$$q(x_{t-1} | x_t, x_0) = q(x_t | x_{t-1}, x_0) \frac{q(x_{t-1} | x_0)}{q(x_t | x_0)} \tag{9}$$

Expand and simplify the three terms at the right end of the equation. The variance $\Sigma_\theta$ of the reverse process can be obtained as a fixed value. Note that [43] indicates that it can also be learned parameters. And the mean of the reverse process $\mu_\theta$ related to $x_t$ and $\tilde{x}_0$:

$$\tilde{\mu}_t(x_t, \tilde{x}_0) = \frac{\sqrt{\bar{\alpha}_t - 1} \beta_t}{1 - \bar{\alpha}_t} \tilde{x}_0 + \frac{\bar{\alpha}_t (1 - \bar{\alpha}_{t-1})}{1 - \bar{\alpha}_t} x_t \tag{10}$$

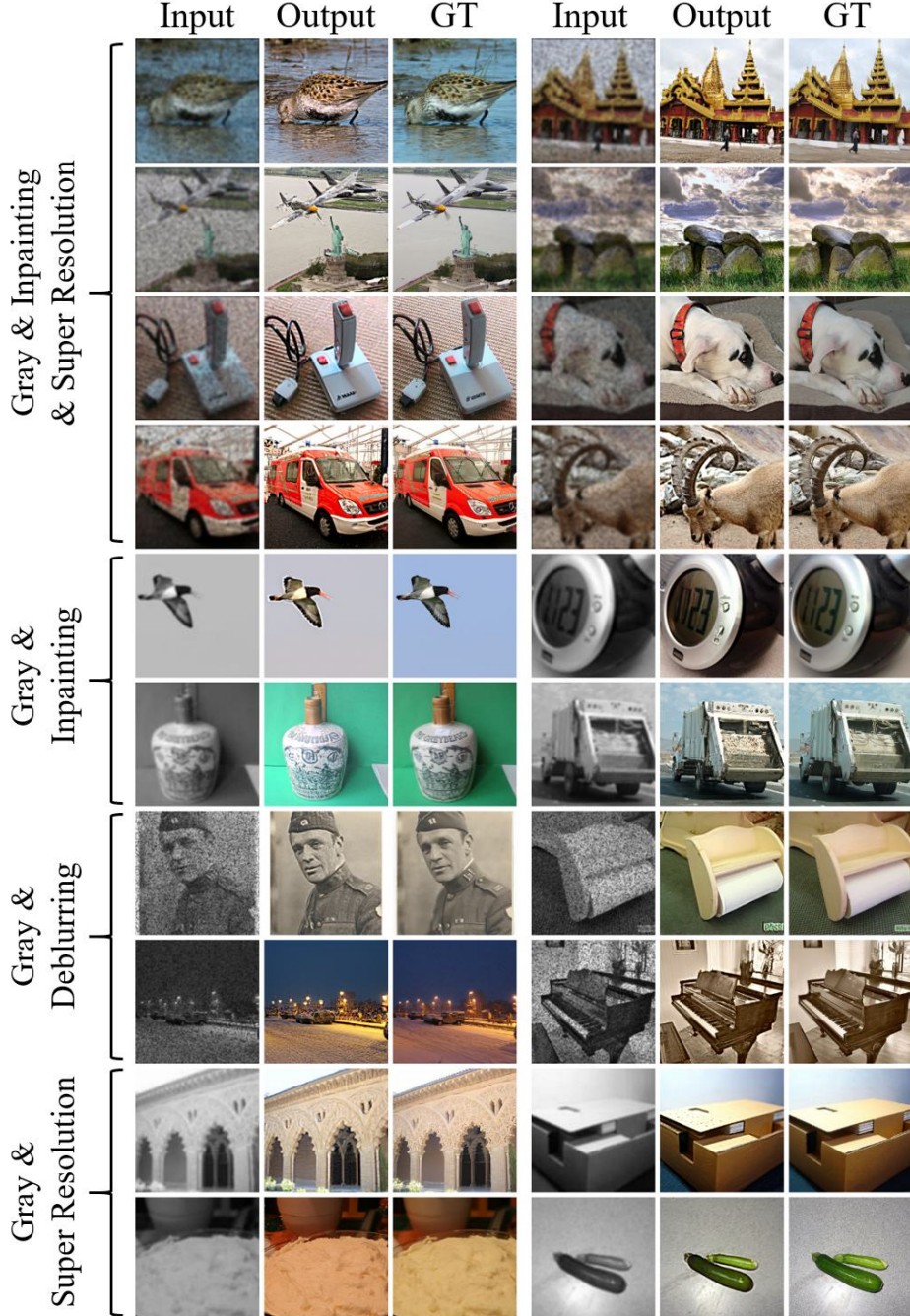

Figure 13: Image results of BIR-D in multi-degradation tasks on the ImageNet dataset. Each row in the figure consists of two sets of images, and the left, middle and right images of each set represent input, output, and ground truth respectively.

According to the formula of the forward process, $\tilde{x}_0$ can be predicted by $x_t$, where $\epsilon$ is a noise function approximator obtained by a neural network $\theta$.

$$\tilde{x}_0 = \frac{x_t}{\sqrt{\bar{\alpha}_t}} - \frac{\sqrt{1-\bar{\alpha}_t}\epsilon_\theta(x_t, t)}{\sqrt{\bar{\alpha}_t}} \tag{11}$$

Substitute it into Equation (10) to obtain the mean value $\mu_\theta$:

$$\mu_\theta(x_t, t) = \frac{1}{\sqrt{\alpha_t}}(x_t - \frac{\beta_t}{\sqrt{1 - \bar{\alpha}_t}}\epsilon_\theta(x_t, t)) \tag{12}$$

## C   Related Work

**Diffusion Model for Image Restoration.**  Image restoration and denoising have seen various advancements with diffusion-based models [16; 17]. They have been thoroughly explored for linear inverse problems [11; 10], nonlinear inverse problems [44; 10]. To alleviate the fixed- and small-size generation of diffusion models, patch-based algorithm [45] and large-size generation [46; 12] are proposed. Our model introduces the guidance of degraded images to form an unconditional diffusion model, and attempts to simulate and update the degradation function in real-time, making it suitable for general tasks while maintaining both image quality and efficiency.

**Blind Image Restoration.** Many problem-solving approaches have emerged in the field of blind image restoration [47; 48]. The emergence of GANs [2; 49] provides several solutions for unsupervised learning in blind image restoration. Generative prior-based image restoration methods [50; 51] employ deep generative models to learn the prior and demonstrate that GAN can be employed as a density estimation model to address various image restoration task. Besides, [51; 52] can also generate a broad spectrum of highly nonlinear complex degradation without any explicit supervision through training in concert with a deep restoration neural network governed by a minmax criterion. On top of GANs and other relevant methods, DDPMs are more studied for this task due to the enhanced diversity. For instance, both DiffBIR [17] and GDP [10] leverage generative diffusion priors for blind image restoration. BlindDPS [53] introduces parallel diffusion models for solving blind inverse problems when the functional forms are known. PromptIR [54] uses prompts to encode degradation-specific information and dynamically guide the recovery of the network. Nevertheless, these methods are still limited to specific tasks. BIR-D can be regarded as a unified solver for multiple restoration tasks by simultaneously estimating the recovered images and specific degradation models.

## D   Adaptive Guidance Scale

In the reverse process of the diffusion model, we added guidance from y to transform the original reverse denoising distribution $p_\theta(x_t \mid x_{t+1})$ into a conditional distribution $p_\theta(x_t \mid x_{t+1}, y)$. This distribution can be further simplified:

$$p_\theta(x_t \mid x_{t+1}, y) = \frac{p_\theta(x_t, x_{t+1}, y)}{p_\theta(x_{t+1}, y)} \tag{13}$$

$$= \frac{p_\theta(x_t, x_{t+1}, y)}{p_\theta(y \mid x_{t+1})p_\theta(x_{t+1})} \tag{14}$$

$$= \frac{p_\theta(x_t \mid x_{t+1})p_\theta(y \mid x_t, x_{t+1})p_\theta(x_{t+1})}{p_\theta(y \mid x_{t+1})p_\theta(x_{t+1})} \tag{15}$$

$$= \frac{p_\theta(y \mid x_t, x_{t+1})p_\theta(x_t \mid x_{t+1})}{p_\theta(y \mid x_{t+1})} \tag{16}$$

$$= \frac{p_\theta(y \mid x_t)p_\theta(x_t \mid x_{t+1})}{p_\theta(y \mid x_{t+1})} \tag{17}$$

$$= \frac{p(y \mid x_t)p_\theta(x_t \mid x_{t+1})}{p_\theta(y \mid x_{t+1})} \tag{18}$$

In this formula, distribution $p_\theta(y \mid x_{t+1})$ is independent of $x_t$, so we use the constant $N$ instead:

$$p_\theta(x_t \mid x_{t+1}, y) = \frac{1}{N}p_\theta(x_t \mid x_{t+1})p_\theta(y \mid x_t) \tag{19}$$

Therefore, compared to the original diffusion model, the conditional reverse process requires approximation of $p_\theta(y \mid x_t)$. We used a heuristic approximation method:

$$p(y \mid x_t) = \frac{1}{N} \exp(-s\mathcal{L}(\mathcal{D}(x_t), y)) \tag{20}$$

Where $\mathcal{L}$ is image distance metric, which represents the MSE loss in this experiment. K is the constant in the above formula, which serves as a normalization factor here. And s is a guidance scale, which is used to control the magnitude of guidance. Take the logarithm of both sides of the equation:

$$\log p(y \mid x_t) = -\log N - s\mathcal{L}(\mathcal{D}(x_t), y) \tag{21}$$

When the diffusion step approaches infinity, $\|\Sigma\| \to 0$, so we can assume that distribution $\log p_\theta(y \mid x_t)$ has low curvature compared to $\Sigma^{-1}$. We can perform Taylor expansion on distribution $\log p_\theta(y \mid x_t)$ around $x = \mu$ and take the first two terms:

$$\log p_\theta(y \mid x_t) \approx \log p(y \mid x_t) \mid_{x_t=\mu} + (x_t - \mu)^T \nabla_{x_t} \log p_\theta(y \mid x_t) \mid_{x_t=\mu} \tag{22}$$

$$= (x_t - \mu)^T g + C \tag{23}$$

Where $g = \nabla_{x_t} \log p_\theta(y \mid x_t) \mid_{x_t=\mu}$, $C = \log p(y \mid x_t) \mid_{x_t=\mu}$. By combining the heuristic approximation formula and Taylor expansion formula mentioned above, we can simplify the empirical formula for the guidance scale:

$$-\log N - s\mathcal{L}(\mathcal{D}(x_t), y) = (x_t - \mu)^T g + C \tag{24}$$

The empirical formula is shown below. For each image at every moment t, the applicable value of the guidance scale can be calculated.

$$s = -\frac{(x_t - \mu)^T g + C + \log N}{\mathcal{L}(\mathcal{D}(x_t), y)} \tag{25}$$

Because here y is a loss image without noise, while $x_t$ itself has noise. The use of MSE errors between $x_t$ and y can lead to the introduction of noise into the guidance process. Therefore, we are using the MSE error between the estimated value of $\tilde{x}_0$ and y here, and the above formula needs to be corrected as:

$$s = -\frac{(x_t - \mu)^T g + C + \log N}{\mathcal{L}(\mathcal{D}(\tilde{x}_0), y)} \tag{26}$$

## E  Multi-Guidance Blind Image Restoration

BIR-D is capable of accepting multiple input images to incorporate multi-guidance during the reverse steps. Taking HDR image restoration task as an example, BIR-D receives three images as inputs separately. As shown in Figure 14 and Algorithm 2, BIR-D uses three degradation functions for three input images. In each sampling step, after obtaining $\tilde{x}_0$, $\tilde{x}_0$ is respectively substituted into three degradation function at diffusion step $t$. The parameters of convolution kernels and masks are updated by measuring the gradient of its parameters with the distance metric. The average of three distance metrics are used as the overall loss to update the mean and variance used during sampling. The empirical formula of adaptive guidance scale is also based on this loss.

## F  The Optimal Size of Optimizable Convolutional Kernel.

In the main paper, in order to assure the versatility of BIR-D, we used convolution kernels of size $7 \times 7$ for all tasks. Nevertheless, for different types of tasks, the size of the convolution kernel might

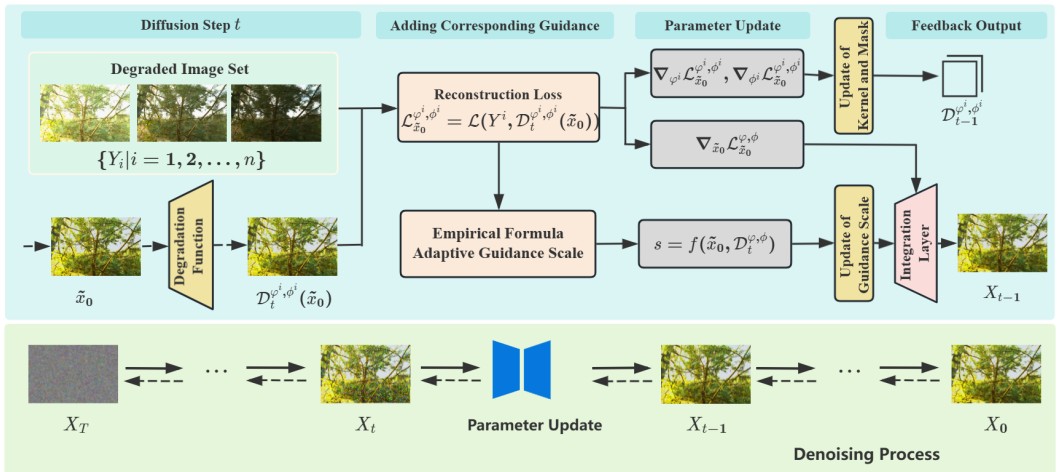

Figure 14: BIR-D image restoration pipeline for multi-guidance tasks.

---

**Algorithm 2:** BIR-D with the multi-guidance of degraded images set $\left\{y^i | i = 1, 2, \ldots, n\right\}$ (For HDR image restoration tasks, n=3), given a diffusion model noise prediction function $\epsilon_\theta(x_t, t)$.

---

**Input:** Degraded image set $\left\{y^i | i = 1, 2, \ldots, n\right\}$. For each image $y^i$ in the set, there is a corresponding degradation function $\mathcal{D}^i$ composed of optimized convolutional kernels $\mathcal{K}^i$ with parameters $\varphi^i$ and mask $\mathcal{M}^i$ with parameters $\phi^i$, learning rate $l$, distant measure $\mathcal{L}^i$.

**Output:** Output image $x_0$ conditioned on set $\left\{y^i | i = 1, 2, \ldots, n\right\}$.

Sample $x_T$ from $\mathcal{N}(0, I)$

**for** t from T to 1 **do**

$\quad \tilde{x}_0 = \frac{x_t}{\sqrt{\bar{\alpha}_t}} - \frac{\sqrt{1-\bar{\alpha}_t}\epsilon_\theta(x_t,t)}{\sqrt{\bar{\alpha}_t}}$

$\quad$ **for** i from 1 to n **do**

$\quad\quad \mathcal{L}_{\varphi^i,\phi^i,\tilde{x}_0} = \mathcal{L}(y^i, \mathcal{D}^{\varphi^i,\phi^i}(\tilde{x}_0))$

$\quad\quad \varphi^i \leftarrow \varphi^i - l\nabla_{\varphi^i}\mathcal{L}_{\varphi^i,\phi^i,\tilde{x}_0}$

$\quad\quad \phi^i \leftarrow \phi^i - l\nabla_{\phi^i}\mathcal{L}_{\varphi^i,\phi^i,\tilde{x}_0}$

$\quad \mathcal{L}_{\varphi,\phi,\tilde{x}_0} = \sum_{i=1}^n \mathcal{L}_{\varphi^i,\phi^i,\tilde{x}_0}$

$\quad s = -\frac{(x_t-\mu)^T g + C + \log N}{\mathcal{L}_{\varphi,\phi,\tilde{x}_0}}$

$\quad \tilde{x}_0 \leftarrow \tilde{x}_0 - \frac{s(1-\bar{\alpha}_t)}{\sqrt{\bar{\alpha}_{t-1}}\beta_t}\nabla_{\tilde{x}_0}\mathcal{L}_{\varphi,\phi,\tilde{x}_0}$

$\quad \tilde{\mu}_t = \frac{\sqrt{\bar{\alpha}_{t-1}}\beta_t}{1-\bar{\alpha}_t}\tilde{x}_0 + \frac{\sqrt{\bar{\alpha}_t}(1-\bar{\alpha}_{t-1})}{1-\bar{\alpha}_t}x_t$

$\quad \tilde{\beta}_t = \frac{1-\bar{\alpha}_{t-1}}{1-\bar{\alpha}_t}\beta_t$

$\quad$ Sample $x_{t-1}$ from $\mathcal{N}(\tilde{\mu}_t, \tilde{\beta}_t I)$

**return** $x_0$

---

| Task | Low-Light Enhancement | | | | | Motion Blur Reduction | |
|---|---|---|---|---|---|---|---|
| | PSNR | SSIM | LOE | FID | PI | PSNR | SSIM |
| kernel size=1 | 13.73 | 0.49 | 118.38 | 78.52 | 5.67 | 31.14 | 0.917 |
| kernel size=3 | 13.90 | 0.54 | 113.89 | 74.41 | 5.24 | 32.07 | 0.937 |
| kernel size=7 | 14.47 | **0.56** | 108.75 | 70.55 | 4.93 | 33.94 | 0.961 |
| BIR-D with $5 \times 5$ kernel | **14.52** | **0.56** | **105.42** | **68.98** | **4.87** | **34.12** | **0.968** |

Table 7: **The ablation study of kernel size in blind issues.**

| Task | $4 \times$ Super resolution | | | | Deblur | | | |
|---|---|---|---|---|---|---|---|---|
| | PSNR | SSIM | Consistency | FID | PSNR | SSIM | Consistency | FID |
| kernel size=13 | 24.05 | 0.66 | 6.65 | 39.02 | 26.12 | 0.74 | 41.29 | 3.09 |
| kernel size=7 | 24.31 | 0.67 | 6.64 | 38.91 | 26.53 | 0.77 | 38.60 | 2.53 |
| kernel size=11 | 24.36 | 0.69 | 6.50 | 38.07 | 26.79 | 0.79 | 38.52 | 2.44 |
| BIR-D with $9 \times 9$ kernel | **24.58** | **0.71** | **6.32** | **37.54** | **27.14** | **0.84** | **37.86** | **2.32** |
| Task | $25\%$ Inpainting | | | | Colorization | | | |
| | PSNR | SSIM | Consistency | FID | PSNR | SSIM | Consistency | FID |
| kernel size=7 | 29.58 | 0.80 | 6.17 | 18.09 | 20.07 | 0.76 | 39.85 | 42.29 |
| kernel size=13 | 31.12 | 0.84 | 5.64 | 16.56 | 21.04 | 0.83 | 37.71 | 38.14 |
| kernel size=11 | 32.91 | 0.86 | 5.41 | 16.17 | 21.57 | 0.85 | 37.69 | 38.01 |
| BIR-D with $9 \times 9$ kernel | **33.59** | **0.90** | **5.18** | **15.73** | **22.09** | **0.89** | **36.12** | **36.58** |

Table 8: **The ablation study of kernel size in linear inverse problem.**

be different. To explore the impact of kernel size on the quality of generated images, we conducted experiments using convolution kernels of different sizes in various types of image restoration tasks. As shown in Table 7, for blind image restoration tasks, the experiment showed that the results of a 5×5 convolution kernel perform best. For linear inverse tasks (Table 8), the optimal convolution kernel size was 9×9.

# G    The Parameter Variations of the Optimizable Mask in the Reverse Steps

In order to visualize the variation of the parameters of convolution kernel mask in the reverse process, we conducted experiments on the test set of the LOL dataset from the low-light enhancement task. As shown in Figure 15, the mask has successfully captured certain intricate details within the low-light images, thereby facilitating the restoration of brightness during the reverse step.

# H    More Blind Image Restoration Results

In this section, we present more generated results for several types of image restoration tasks in blind issues, including low-light enhancement, motion blur reduction, and HDR image restoration.

For the blind face restoration task, we randomly selected 1000 images from the LFW [14] and WIDER [15] test sets as input samples. More image results are shown in Figure 16. BIR-D effectively eliminates the blurring in degraded images, resulting in clearer and more detailed facial images.

For the low-light enhancement task, we use datasets including LOL [22] and VE-LOL-L [23]. The image results are shown in Figure 17. BIR-D achieved excellent results on different datasets, with the brightness of the generated images very close to the ground truth without losing any details during the restoration process.

For the motion blur reduction task, the datasets used include GoPro dataset [30] and HIDE dataset [31]. Figure 18 shows the results of BIR-D in eliminating motion blur. This also demonstrates the capabilities of BIR-D for unknown degradation functions.

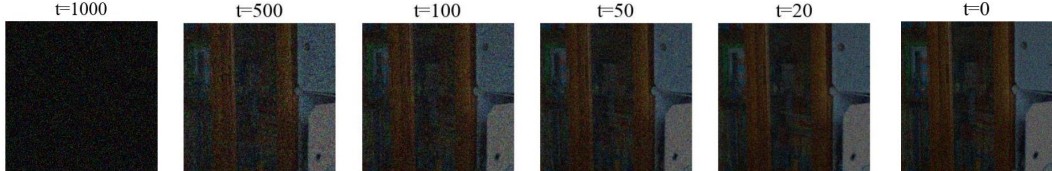

Figure 15: The changing of degradation mask during the sampling process in low light enhancement.

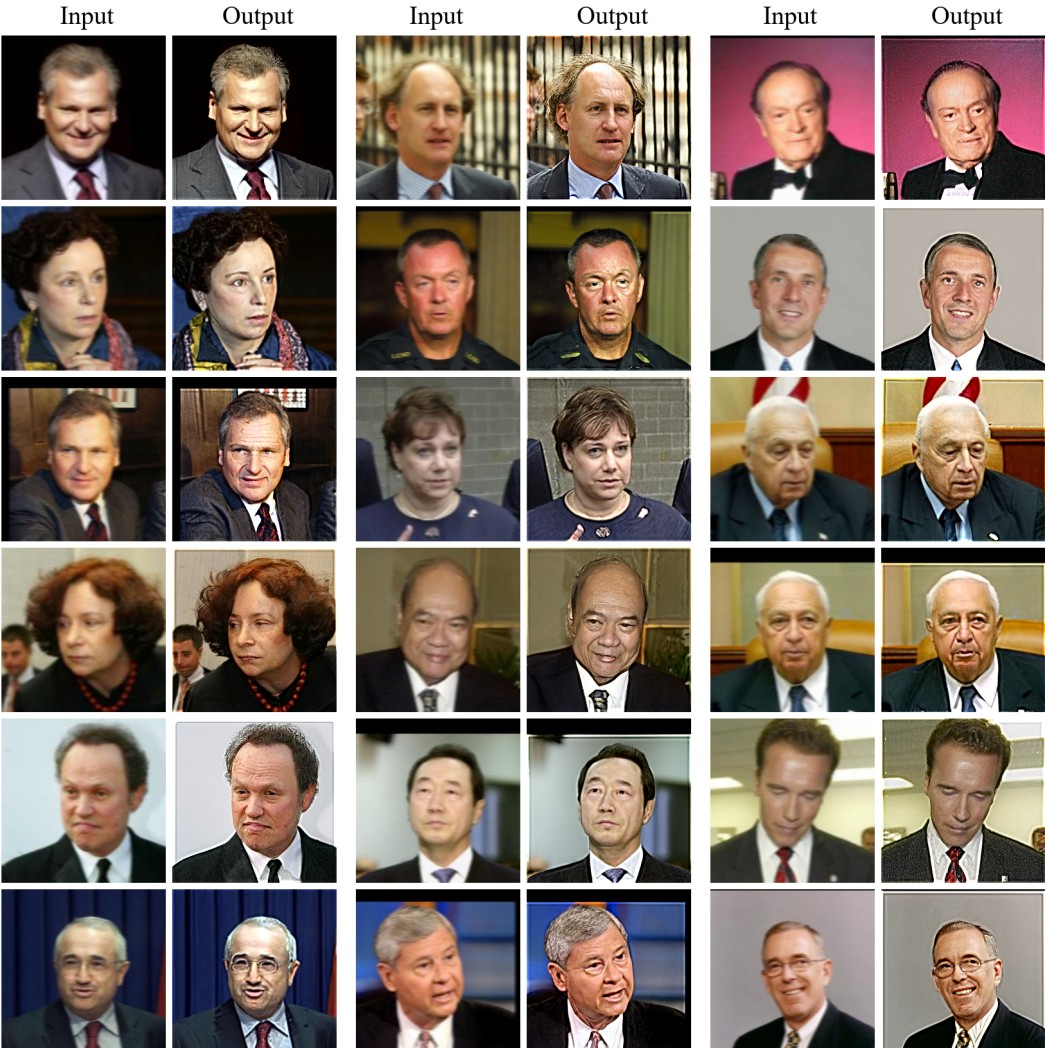

Figure 16: More image generation results for blind face restoration task. Each row consists of three sets of images, with the left and right images representing blurred facial images and BIR-D output images, respectively.

Meanwhile, we use the NTIRE2021 Multi-Frame HDR Challenge dataset [29] to test the HDR image restoration capability of BIR-D. Each image scene contains three LDR images, including long, medium, and short exposures. Following [10], we leverage three images as condition for HDR recovery. The more generated results of this task are shown in Figure 19. The over-exposure part in the image has been corrected, while the low-light part has been enhanced in brightness, resulting in a more distinct and detailed generation result.

| Input | Output | GT | Input | Output | GT |
|-------|--------|-----|-------|--------|-----|

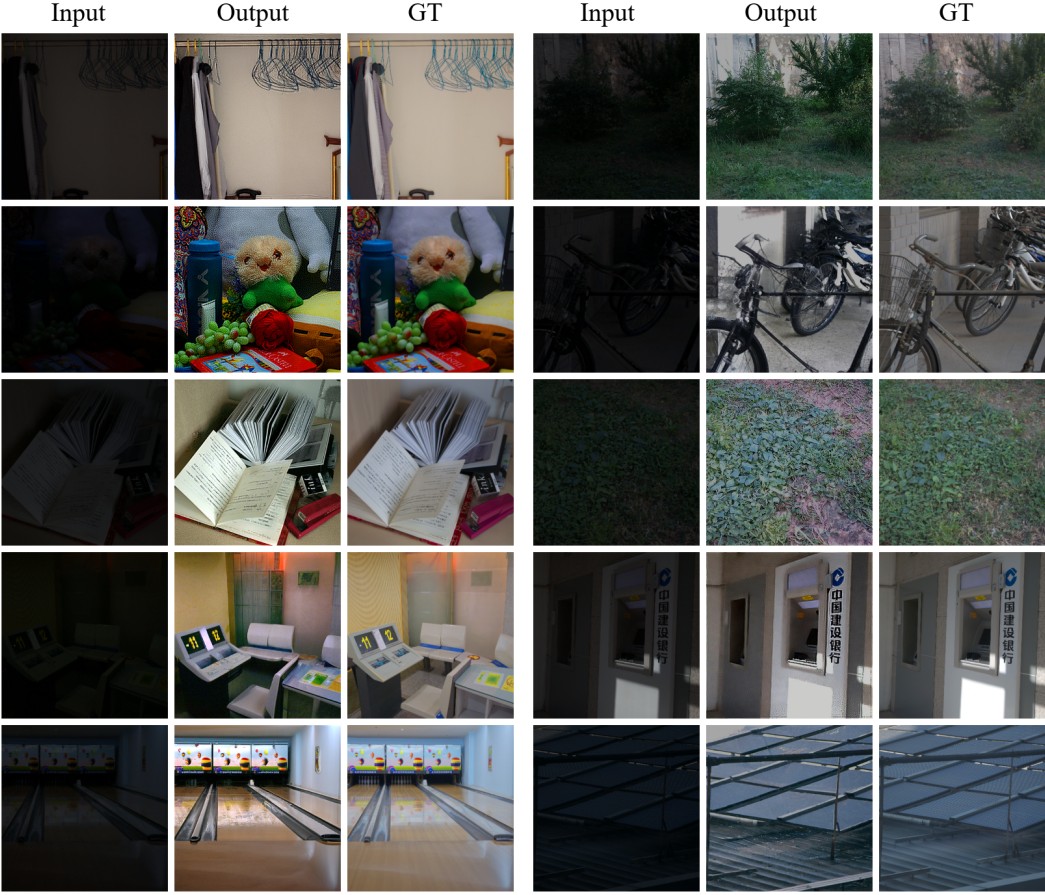

Figure 17: More image generation results for low-light enhancement task. The left half shows the image restoration results of the LOL dataset, and the right half shows the image restoration results of the VE-LOL dataset. The left, middle, and right of each group of images represent the input, BIR-D output, and ground truth respectively.

# I   More Results on Commonly-used Tasks

In this section, we will provide more image restoration results on commonly used tasks, including super-resolution, colorization, deblurring, and inpainting tasks. Noted that although BIR-D has the capability to perform image restoration when the diffusion function is unknown, in order to ensure a fair comparison with other methods, an initially set and known degradation function will be used, and all image generation results and comparison metrics will be obtained under this condition. The images used in the test are all from the ImageNet dataset, and the degradation method and parameters of the input images are the same as the state-of-the-art methods, including DGP [18], SNIPS [20], DDRM [11], GDP [10], which ensures the effectiveness and fairness of the comparison.

As shown in Figure 20, BIR-D effectively removes the blur in the input image and effectively restores the details in the image. For the inpainting task, we set 25% of pixels in images to have missing pixel values, and use this as a degraded image to test the image restoration ability of BIR-D. Figure 21 shows that BIR-D has the ability to restore these 25% of missing pixels, resulting in an overall output image that is closer to the ground truth. Meanwhile, we set the ratio to 4 in the super-resolution task. As shown in Figure 22, low-resolution images can be restored from BIR-D to high-resolution images without losing clarity, while preserving various subtle details of natural images. Besides, for a given grayscale image, Figure 23 shows the level of color restoration by BIR-D for the image, and the output results also indicate the ability of BIR-D to solve the colorization task.

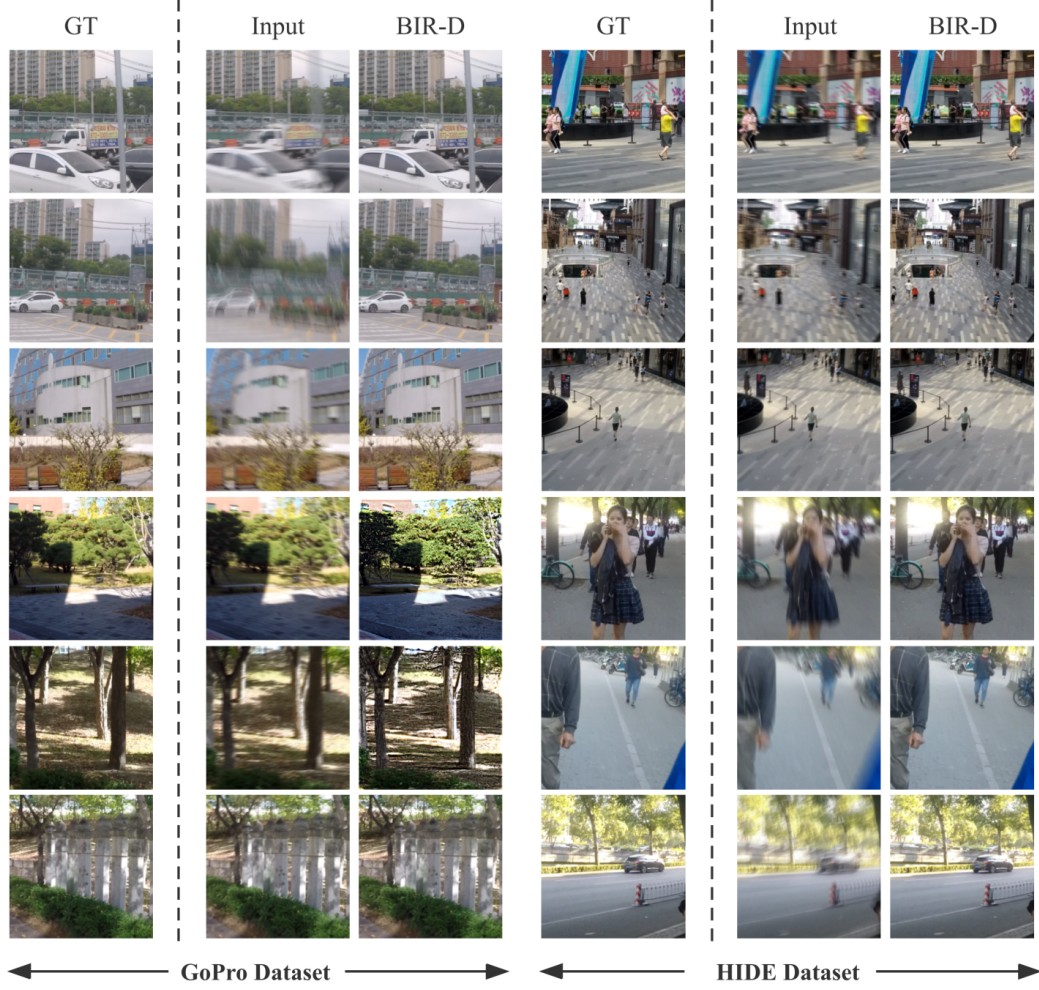

| GT | Input | BIR-D | GT | Input | BIR-D |

Figure 18: More image restoration results of motion blur reduction task on GoPro dataset and HIDE dataset.

## J   Limitation and Future Works

Since we utilize a single pre-trained unconditional diffusion model provided by [13], the generation speed will increase when dealing with image size increasing due to the patch-based solution. In the future, it is a very promising direction to use stable diffusion to achieve faster speed in large-size image restoration.

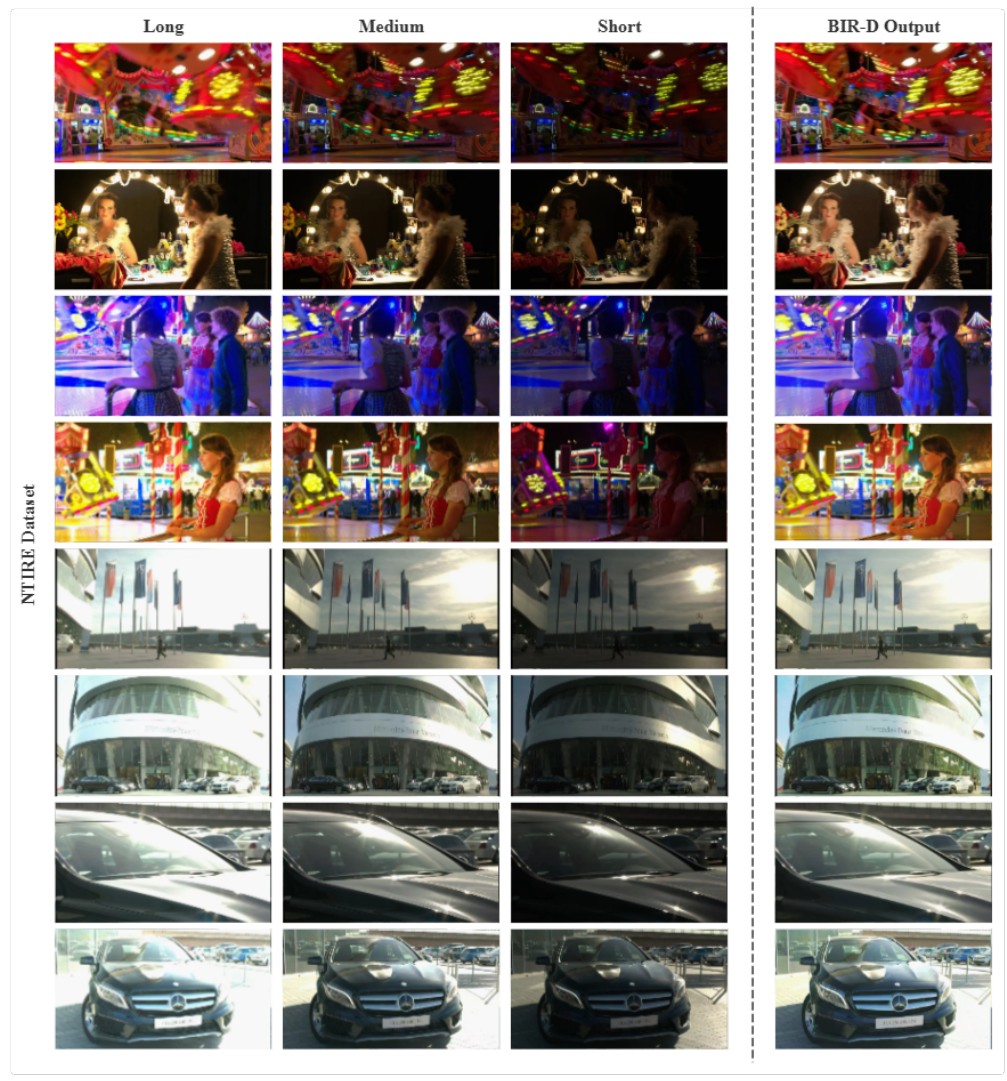

Figure 19: More image restoration results of HDR image recovery task on NTIRE2021 Multi-Frame HDR Challenge dataset.

Input Pre-training Output GT  Input Pre-training Output GT

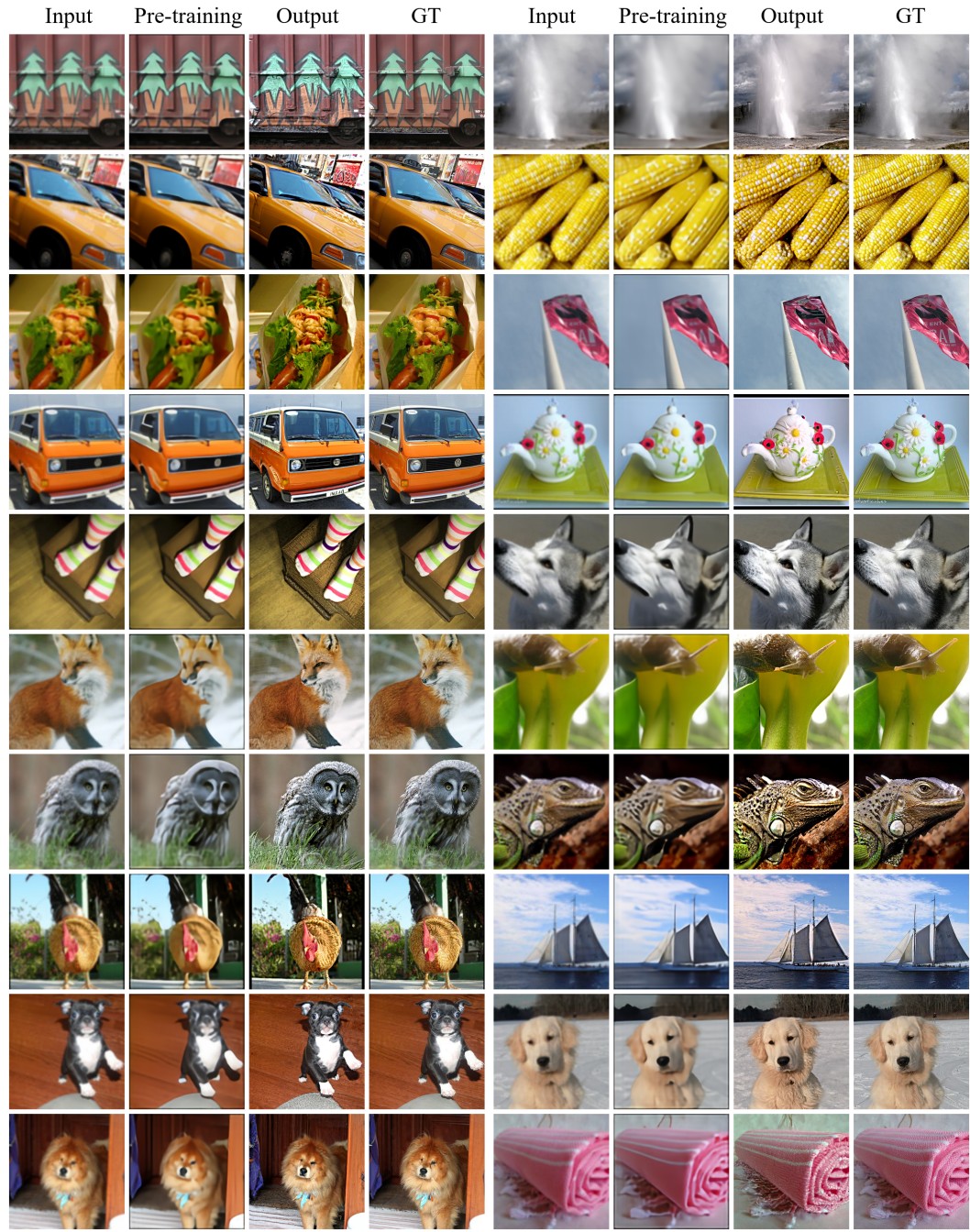

Figure 20: The image generation result of the deblurring task, where each horizontal row is composed of two sets of images, each set of images representing the input image, the image after pre-training model, the output image of BIR-D, and the ground truth from left to right.

Input        Output        GT        Input        Output        GT

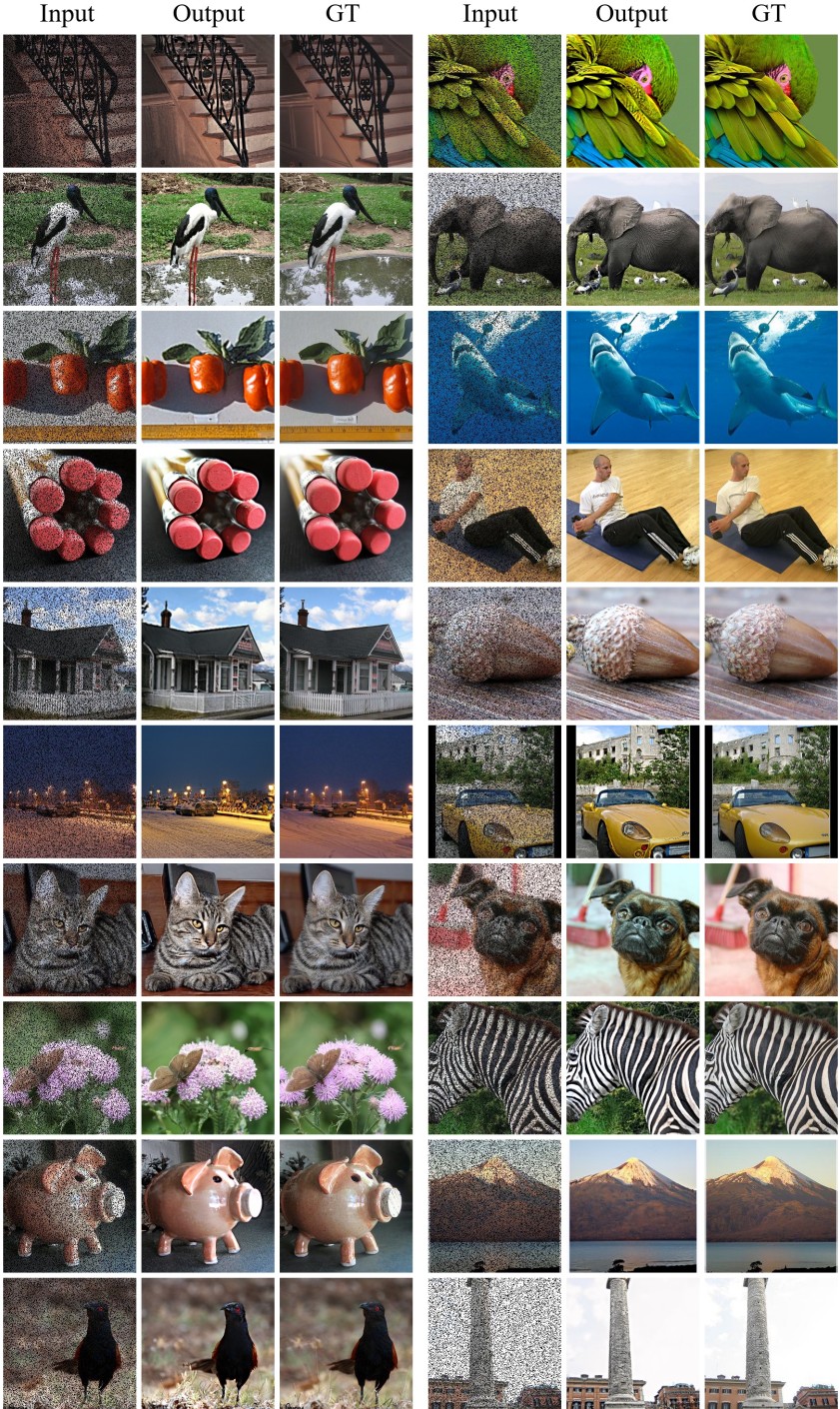

Figure 21: The image generation result of the inpainting task, where each row is composed of two sets of images and the left, middle, and right images of each set represents the degraded image, the output image of BIR-D, and the ground truth, respectively.

Input       Output       GT          Input       Output       GT

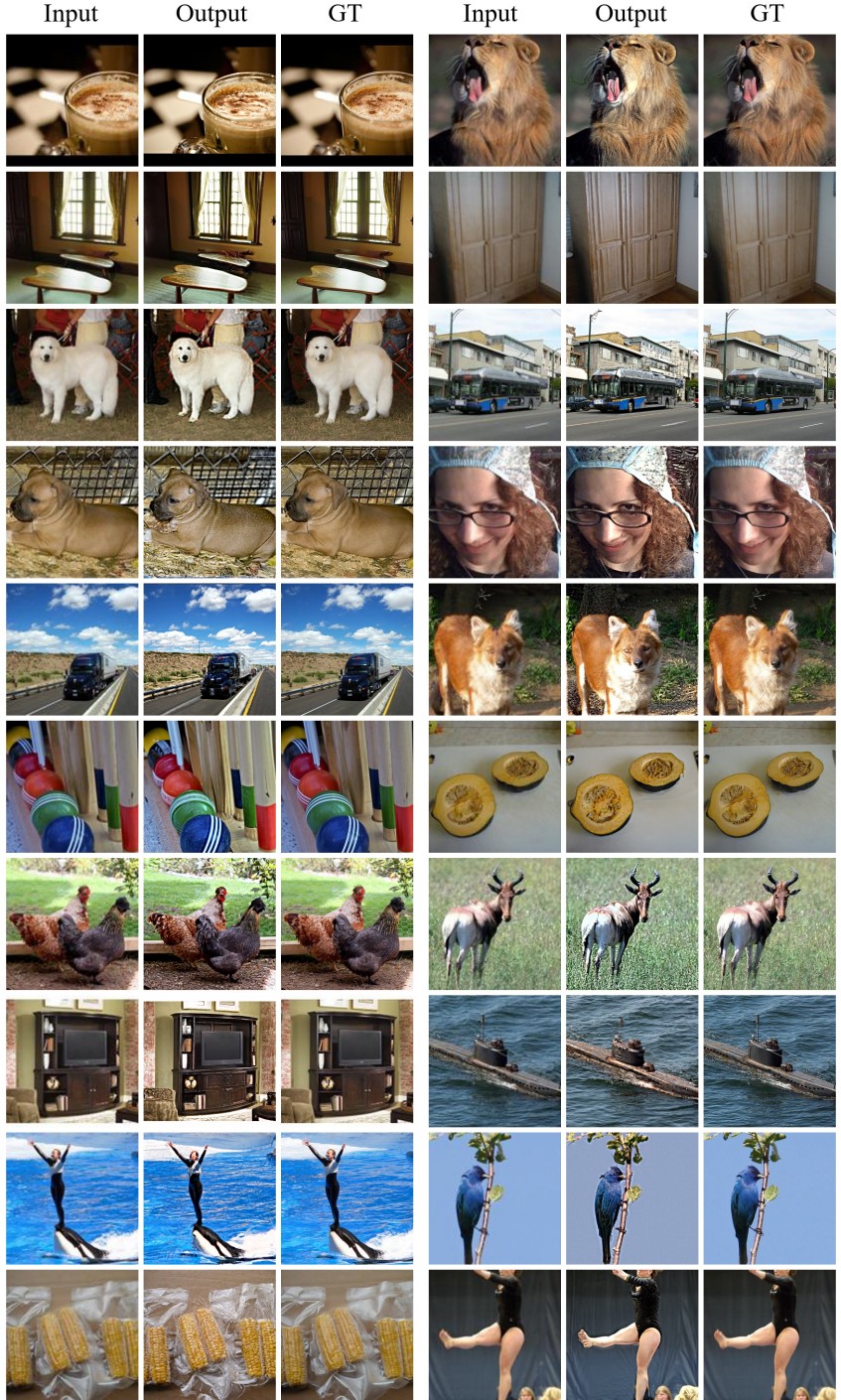

Figure 22: The image generation result of the 4 × super-resolution task, the left, middle, and right images of each set represent the low-resolution images processed by the resize function, the output images of BIR-D and the ground truth respectively.

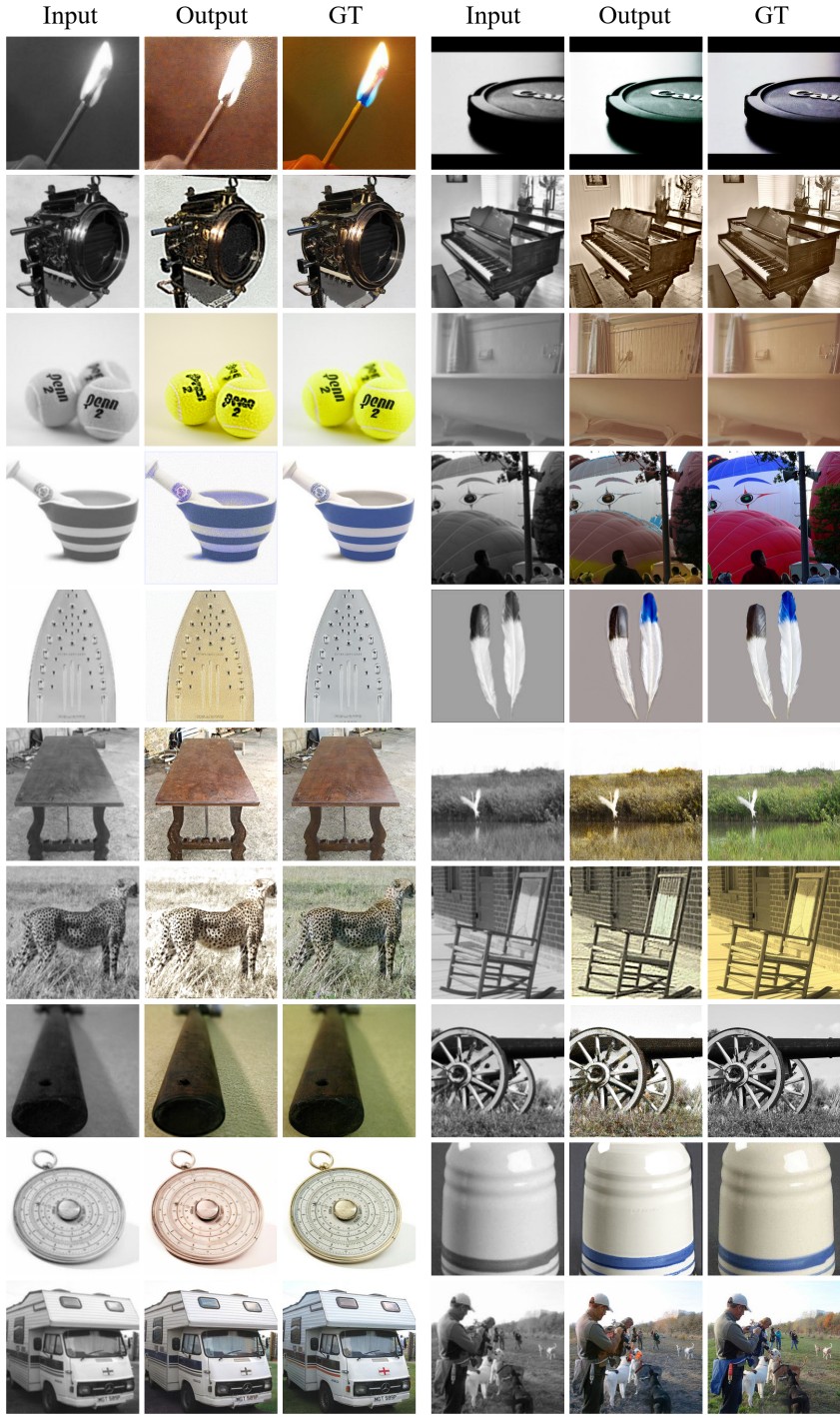

Figure 23: The image generation result of the colorization task, the left, middle and right images of each set represent the grayscale images, the output images of BIR-D and the ground truth respectively.

