# OpenReview forum: "Taming Generative Diffusion Prior for Universal Blind Image Restoration"
_NeurIPS.cc/2024/Conference — NeurIPS 2024 poster_

### Official Review · Reviewer_rhDr · 2024-07-06

**Soundness:** 2
**Presentation:** 2
**Contribution:** 2
**Rating:** 3
**Confidence:** 5

**Summary:**

This paper proposes a blind image restoration method by using a pre-trained diffusion model without additional prior knowledge. The proposed adaptive guidance scale is fancy which uses the loss function to judge its value while the degradation function design is confusing. The results on real-world benchmarks show great performance success, but the experiment to validate their proposed modules is fragile.

**Strengths:**

1. The author proposes a non-training strategy to handle the blind image restoration tasks with no additional prior knowledge and achieve SOTA performance in the real world.
2. The way to control the guidance scale is fancy.

**Weaknesses:**

1. The introduction of the method section is confusing, first, what is the “$\sum$” in Line 112 and Line 421? Second, the design of the degradation function D does not make sense, why does adding the M term estimate the noises and what does the noise mean here (Line 72)?
2. The proposed method is unreliable, the author said that they do not have additional prior information, but in my view, the usage of the pre-trained model (Diff-BIR) is the prior knowledge, this paper is just an extra refiner to refine a coarse-clean image to a better one. To validate the proposed method, do the extra ablation study on all the benchmarks without pre-trained model and show the quantitative and qualitative results w, w/o it.
3. The ablation study is fragile. i) Visualize the result of the degradation function D for degrading the clean image. ii) Visualize what M learned for different tasks. iii) Show the value changing of guidance scales during the denoising stage and propose the theoretical analysis of the changing trend.

**Questions:**

See the weakness.

**Limitations:**

Yes.

---

> ### Author Rebuttal · Authors · 2024-08-06
>
> We appreciate the valuable questions and suggestions raised by Reviewer rhDr regarding this article.
>
> > **Q1:** "(i)What is the “∑” in Line 112 and Line 421? (ii)The design of the degradation function D does not make sense, why does adding the M term estimate the noises and what does the noise mean here (Line 72)?"
>
> **A:** Thanks for your question.
>
> * For question (i), $\Sigma$ appearing in lines 112 and 421 both represent the variance of the unconditional distribution of the reverse process of the diffusion model in this paper. As indicated in line 112 and line 13 of Algorithm 1 ($\tilde{\beta}_t=\frac{1-\bar{\alpha}\_{t-1}}{1-\bar{\alpha}_t}\beta_t$, $\Sigma=\tilde{\beta}_t I$), since its parameters are all known, "$\Sigma$" is a constant.
>
> * For question (ii), in some blind image restoration tasks, such as low-light enhancement tasks, the degree of noise contained in different subregions of the image during the sampling process varies. Using only a single convolution kernel cannot effectively restore regions with strong noises. So we have added a mask $\mathcal{M}$ with the same size as the degraded image here to effectively simulate different noises in local regions of images. As shown in Global PDF Figure 4 and 5, taking the HDR image recovery and low-light enhancement task as an example, the degradation mask helps the model to achieve image restoration of local regions with significant brightness differences. The degradation mask also learned the detailed information of various regions in the image.
>
> ___
>
> > **Q2:** (i)"The usage of the pre-trained model is the prior knowledge, this paper is just an extra refiner to refine a coarse-clean image to a better one."
> (ii)"To validate the proposed method, do the extra ablation study on all the benchmarks without pre-trained model and show the quantitative and qualitative results w, w/o it."
>
> **A:** We thank the reviewer for the comment.
> * For question (i), we introduce the first stage pre-training model [R1] here to improve the blind image restoration performance of the model. The first stage pre-training model is able to provide a better initial state of images for our BIR-D. A more detailed analysis of this effect was conducted in the ablation study. It is worth noting that we do not use the pre-training diffusion model of DiffBIR, but rather its first stage pre-training model. We only introduce this pre-training model in the deblur and motion blur reduction tasks, and we have made modifications in the revised paper to address any potential misunderstandings caused by the original description. When there is a significant deviation between the initial value of the model degradation function and the actual degradation, the pre-training model can improve its restoration performance. But for general cases, the ablation study validates that the model can generate restoration images with rich detailed information without the need for pre-training models.
> * For question (ii), we have compared the quantitative results with and without the addition of the pre-training model in the ablation study of the main paper and compared the qualitative results in Figure 15 of Appendix G. We only add pre-training models in the tasks of deblurring and motion blur reduction, and the rest of the experimental results are obtained without pre-training models.
>
> ___
>
> > **Q3:** "The ablation study is fragile. (i) Visualize the result of the degradation function D for degrading the clean image. (ii) Visualize what M learned for different tasks. (iii) Show the value changing of guidance scales during the denoising stage and propose the theoretical analysis of the changing trend."
>
> **A:**
>
> * For questions (i) and (ii), Global PDF Figures 1 and 2 show the parameter changes of the degradation function of BIR-D in the LOL dataset of the low-light enhancement task. The convolutional kernel and mask $\mathcal{M}$ of the degradation model both have an upward trend from their initial values, making the overall degradation function approach the true degradation. As shown in global PDF Figure 4 and 5, during the sampling process, the degradation mask learns the detailed information of the image, including local regions with significant brightness differences. This process is obtained by updating the gradient of the distance metric with respect to the degradation mask parameters. The degradation function is composed of an optimizable convolutional kernel and a mask. The mask and the degraded image have the same dimension, which helps to solve the image restoration of local regions with large brightness changes in the image.
> * For question (iii), as shown in Global PDF Figure 3, the guidance scale gradually decreases with the reverse process, which is consistent with the actual situation. The total number of time steps T=1000, after 500 time steps, the difference between $x_t $ and $x_{t-1}$ gradually decreases. That is, the noise simulated in each step gradually decreases. Therefore, the degree of guidance required for each time step should also be correspondingly reduced. At this point, the required guidance scale value also decreases accordingly. Empirical formulas can also provide a reasonable explanation for this trend. When there are time steps $t$<500, the gradient term $g$ also decreases as the degree of change of $x_t$ gradually decreases at each step. And the speed at which the gradient term decreases is greater than the speed at which the distance metric decreases, resulting in a decrease in the guidance scale value (Please see Eq. (3) in the main paper). Compared with the setting of fixed guidance scales in GDP, using adaptive guidance scales in the sampling process is in line with practical requirements. The superior performance in the ablation study in the main paper also demonstrates the advantages of the "adaptive guidance scale".
>
> ___
>
> **Reference:**
>
> [R1] Xinqi Lin et al. "Diffbir: Towards blind image restoration with generative diffusion prior." arXiv, 2023.

---

> > ### Comment · Reviewer_rhDr · 2024-08-12
> >
> > The author only addresses partial of my concern.
> > For the noise mask, the visualization is hard to understand and the explanation is not convincing.
> > For pretrained weight, as you are doing the setting of universal, how can you add it to some of the tasks? This makes me question the veracity of the author's experiment in UNIVERSAL。
> > All in all, the original paper lacks too many experiments and I do not think it can be refined directly.
> > I will keep my rating.

---

> > > ### Author Response · Authors · 2024-08-13
> > >
> > > Dear Reviewer rhDr:
> > >
> > > We sincerely appreciate your time and effort in reviewing our paper. We would like to clarify the following issues you mentioned.
> > >
> > > 1. The setting of mask $\mathcal{M}$ is mainly used to solve the image restoration task of local regions with significant differences in brightness, as using an optimizable convolution kernel alone may not be able to effectively solve the brightness correction task of such regions. As shown in Global PDF Figures 4 and 5, a mask with the same dimension as the degraded image can learn the brightness and detail information of each local region of the image, which also assists the optimizable convolution kernel in simulating the degradation function.
> > >
> > > 2. The first stage pre-training model is only used to improve model performance in the two tasks of deblurring and motion blur reduction. Without the first stage pre-training model, our BIR-D can still achieve image restoration of deblurring and motion blur reduction tasks (see Table 6 in the main text and Figure 15 of Appendix G). It is worth noting that this first stage pre-training model is not used in the other blind image restoration tasks since other tasks can be effectively modeled by our devised optimizable convolution kernel. The universal capacity of our BIR-D comes from the design of an optimizable convolution kernel, which can effectively simulate any degradation models of most blind image restoration tasks. The experiment we conducted also proved this contribution. Here is a summary of the experiment we conducted.
> > >
> > >
> > > | Category       | Task                        | Dataset               | Figure    | Table |
> > > |----------------|-----------------------------|-----------------------|-----------|-------|
> > > |                | Deblurring                  |                       | 1,5(b),15 | 2     |
> > > |                | Colorization                |                       | 1,4,18    | 2     |
> > > | Linear Inverse | Super-resolution            | ImageNet 1k           | 1,5(a),17 | 2     |
> > > |                | Inpainting                  |                       | 1,5(c),16 | 2     |
> > > |                | Multi-task                  |                       | 1,9,10    | -     |
> > > |----------------|-----------------------------|-----------------------|-----------|-------|
> > > |                | BIR in Real-world Dataset   | LFW,Wider             | 1,3,11    | 1     |
> > > |                | Low-light Enhancement       | LOL,VE-LOL,LoLi-Phone | 1,6,12    | 3     |
> > > | Non-linear     | Motion Blur Reduction       | Gopro,HIDE            | 1,8,13    | 4     |
> > > |                | HDR Image Recovery          | NTIRE 2021            | 1,7,14    | 4     |
> > > |                | Realistic Image Restoration | Website               | 1         | -     |
> > >
> > >
> > > ___
> > >
> > > We hope these clarifications will enhance your comprehension of our paper. If you have any further comments, please do not hesitate to mention it. We look forward to further communicating with you.
> > >
> > > Best wishes,
> > >
> > > The Authors

---

> ### Author Response · Authors · 2024-08-12
> **Looking forward to discussion**
>
> Dear Reviewer rhDr:
>
> We sincerely thank you for taking the time to this review and providing valuable comments.
>
> ___
>
> Based on the reviewers' comments, we have made revisions to our manuscript to include the following changes.
>
> * We provide the trends of parameters in the adaptive guidance scale and optimizable convolution kernel in the sampling process to clarify how these designs contribute to the performance of BIR-D.
> * We clarify that we only use the first stage pre-training model for better initialization, rather than the pre-training diffusion model of DiffBIR.  And we analyze the improvement of BIR-D using this first-stage pre-training model.
> * We have supplemented more details on the visualization, description and explanation of some symbols the reviewer mentioned in the review.
>
> ___
>
> We hope our explanations have addressed your concerns. As we are in the discussion phase, we welcome any additional comments or questions regarding our response or the main paper. If further clarification is needed, please do not hesitate to mention it, and we will promptly address your inquiries. We look forward to receiving your feedback.
>
> Best wishes,
>
> The Authors

---

> ### Author Response · Authors · 2024-08-14
>
> Dear reviewer rhDr,
>
> We sincerely thank you for your valuable time and feedback. We hope our existing rebuttal and official comments could address your previous concerns well. As the discussion phase is nearing its end, we remain open to addressing any remaining questions or concerns. If you have any further questions during the next discussion period, please let us know, and we will be happy to answer them. We look forward to receiving your feedback. Thank you once again!
>
> Sincerely,
>
> Best regards,
>
> The Authors

---

### Official Review · Reviewer_CUUT · 2024-07-12

**Soundness:** 2
**Presentation:** 3
**Contribution:** 3
**Rating:** 5
**Confidence:** 4

**Summary:**

This research introduces BIR-D, a novel approach to the universal challenge of blind image restoration. It leverages an adaptable convolutional kernel designed to emulate the degradation model, with the capability to refine its parameters progressively during the diffusion process. Furthermore, the work presents an empirical formula to guide the selection of the adaptive scale, a critical component in enhancing restoration accuracy. Extensive experiments substantiate the method's exceptional performance across a spectrum of restoration tasks, showcasing its robustness and efficacy.

**Strengths:**

This research offers a novel perspective on Classifier-Guidance, highlighting the essential role of the guidance scale in the fidelity of image generation, and points out that applying a fixed guidance scale across all denoising steps is far from ideal. Therefore, it is necessary to innovate a method that enables the adaptive, real-time adjustment of the guidance scale at each stage of the diffusion process for degraded images in specific restoration tasks. The paper presents a robust validation of the BIR-D method through a comprehensive set of experiments across multiple image restoration tasks, such as deblurring, super-resolution enhancement, low light image enhancement, HDR image recovery, and multi-degradation image restoration.

**Weaknesses:**

1. The contribution in question, which utilizes an optimizable convolutional kernel to simulate the degradation model and dynamically update the parameters of the kernel during the diffusion steps, may be perceived as lacking in novelty. You should provide a detailed comparison with the referenced [10], "Generative Diffusion Prior for Unified Image Restoration and Enhancement," about the different strategy for updating the degradation model.
2. The paper's exploration of the 'optimizable convolutional kernel' and 'adaptive guidance scale' could be enhanced by including an analysis of convergence trends or parameter behavior over time. Such analyses would clarify how these elements contribute to the method's performance.
3. The notations in this paper may lead to misunderstandings. Specifically, in Formula (6), the representation of $g$ lacks the subscript $xt=μ$, which is critical for clarity. Furthermore, the $N$ in Formula (19) should be distinguished from the $N$ used in Formula (20) to avoid ambiguity. Additionally, on line 418, the symbol $K$ should be replaced with $N$ for consistency.
4. The paper's explanation is not sufficiently clear, such as how BIR-D can accomplish multi-guidance blind image restoration.

**Questions:**

1. Upon my review, the formula for calculating the guidance scale $s$ in equation (3), once combined and simplified with equation (1), yields an identity. This suggests that the information obtained from the current $xt$ sampling is independent of the update to $s$. Could you clarify this issue?

---

> ### Author Rebuttal · Authors · 2024-08-06
>
> We greatly appreciate the valuable comments and suggestions provided by Reviewer CUUT on this article.
>
> > **Q1:** Detailed comparison with GDP.
>
> **A:** We greatly appreciate your suggestions. BIR-D has the following differences compared to GDP.
>
> 1. Differences in the setting of degradation function.
> * In linear inverse image restoration tasks, GDP needs to be given a degradation function as well as the initial value, and the degradation function remains unchanged in the sampling process. In contrast, the optimizable convolutional kernel in BIR-D effectively circumvents this issue.
> * For blind image restoration tasks, GDP assumes a degradation form of $Y=fX+\mathcal{M}$. But this setting is only effective for specific tasks such as low-light enhancement and HDR recovery, as using only $f$ as a coefficient cannot simulate more complex degradation scenarios. By contrast, the use of optimizable convolutional kernels and masks in BIR-D can simulate more unknown degradation.
> * GDP is only able to restore images with two degradations. In contrast, BIR-D can handle more complex scenarios involving 3-4 types of mixed degradation by utilizing optimizable convolution kernels to simulate degradations, offering greater flexibility.
>
> 2. Differences in the setting of the guidance scale.
> * The guidance scale in GDP is manually grid-searched and set for various types of degradation. For images from the dataset or denoised images in every sampling step, the guidance scale remains unchanged. If the guidance scale is not correctly set, there are mineral-like textures in the image results (Global PDF Figures 6). BIR-D's adaptive guidance scale resolves this issue by dynamically calculating the optimal guidance scale throughout the sampling process, showcasing its versatility.
> ___
>
> > **Q2:** "Enhanced analysis of convergence trends or parameter behavior of the 'optimizable convolutional kernel' and 'adaptive guidance scale' over time."
>
> **A:** Thank you for your valuable suggestion. To answer this question, we performed additional experiments on the test set of the LOL dataset from the low-light enhancement task.
>
> * For the "optimizable convolutional kernel", the mean of optimizable convolution kernel parameters increases with the sampling process (Global PDF Figure 1). This increase in magnitude is influenced by the gradient of the distance metric with respect to the parameter. When the sampling step $t$<500, the difference between $\tilde{x}_0$ and $y$ changes minimally, resulting in correspondingly smaller gradient values. Therefore, the parameters of the convolutional kernel gradually converge towards the actual degradation function.
> * For the "adaptive guidance scale", the guidance scale gradually decreases with the sampling step (Global PDF Figure 3), which aligns with the actual situation. When the sampling step $t$<500, the difference between $x_t$ and $x_{t-1}$ diminishes as $t$ decreases, indicating a reduction in the simulated noise at each step. Therefore, the level of guidance required for each sampling step should also be reduced accordingly, leading to a decrease in the required guidance scale values.  According to Eq. 3 in the main paper, when $t$ is small, the gradient term $g$ also decreases due to the small change in $x_t$ at each step. The speed of the gradient term decreases is greater than the speed of the distance metric decreases, resulting in a decrease in the value of the guidance scale.
> ___
>
> > **Q3:** "(i)In Formula (6), the representation of g lacks the subscript xt=μ. (ii)The N in Formula (19) should be distinguished from the N used in Formula (20) to avoid ambiguity. (iii)On line 418, the symbol $\mathcal{K}$ should be replaced with N for consistency."
>
> **A:** We thank the reviewer for pointing this typo out.
> * For question (i), thank you for reminding us that the subscript xt=μ was overlooked here. We have revised the description of $g$ in the revised version.
> * For questions (ii) and (iii), we have addressed this issue in the revised paper. In the paper, $\mathcal{K}$ refers to the optimized convolutional kernel used in the model and N refers to the constant $p_\theta(y|x_{t+1})$.
>
> ___
>
> > **Q4:** The explanation of accomplishing multi-guidance blind image restoration.
>
> **A:**
> Thank you for pointing out this question. Taking HDR image restoration task as an example, BIR-D receives three images as inputs separately. As shown in Global PDF Figures 7 and Algorithm 1, BIR-D uses three degradation functions for three input images. In each sampling step, after obtaining $\tilde{x}_0$, $\tilde{x}_0$ respectively go through into three degradation functions at sampling step $t$. The parameters of convolution kernels and masks are updated by measuring the gradient of their parameters with the distance metric. The average of three distance metrics is used as the overall loss to update the mean and variance used during sampling. The empirical formula of the adaptive guidance scale is also based on this loss. We will incorporate figure, algorithm, and corresponding explanations into the future version to make the introduction of multi-guidance clearer.
>
> ___
>
> > **Q5:** "The formula for calculating the guidance scale s in Eq. (3), once combined and simplified with Eq. (1), yields an identity. This suggests that the information obtained from the current xt sampling is independent of the update to s."
>
> **A:** Thanks for your question. The $s$ comes from heuristic algorithms (Eq. 1), so we attempt to approximate the quantity $\log{}{p(y|x_t)}$ on the left-hand side of Eq. 1. We conducted Taylor expansion on it around $x_t=\mu$ (the rationality analysis of the expansion is located in Appendix D). By combining heuristic algorithm (Eq. 1) with Taylor expansion (Eq. 23), an empirical formula for guidance scale can be obtained. The update of guidance scale is mainly used for updating $\tilde{x}_0$, and the updated  $\tilde{x}_0$ is used to sample $x\_{t-1}$.

---

> > ### Comment · Reviewer_CUUT · 2024-08-14
> >
> > Most of the concerns have been addressed in the authors' response. I will raise the score.

---

> > > ### Author Response · Authors · 2024-08-14
> > >
> > > Dear Reviewer CUUT:
> > >
> > > We sincerely appreciate your helpful and constructive review and are pleased to see your decision to raise your score. Based on your valuable suggestions, we will provide detailed explanations of the differences between BIR-D and GDP in the future version to highlight the strengths and improvements of BIR-D. Meanwhile, the parameter trends of the kernel and mask will be incorporated into our future version. We will also integrate the pipeline of multi-guidance BIR-D in the future version to make our multi-guidance method clearer. To further support our paper, we will carefully release our code. Thank you once again for your recognition of our work and the valuable time you have invested in this review.
> > >
> > > Best regards,
> > >
> > > The Authors

---

> ### Author Response · Authors · 2024-08-12
> **Looking forward to discussion**
>
> Dear Reviewer CUUT:
>
> We sincerely thank you for devoting time to this review and providing valuable comments.
>
> ___
>
> Based on the reviewers' comments, we have made revisions to our manuscript in the following areas.
>
> * We have supplemented the trends of parameters in the adaptive guidance scale and optimizable convolution kernel in the sampling process to better clarify how these designs contribute to the BIR-D's performance.
> * We have listed and analyzed the advantages and improvements of BIR-D compared to GDP from various perspectives. Importantly, we have re-clarified that the challenges previously associated with GDP are effectively addressed by BIR-D.
> * We have provided more details on multi-guidance blind image restoration, including diagram and pseudocode in Global PDF.
> * We have clarified the necessity of proposing empirical formulas with guidance scales and optimized the derivation approach and process to make it clearer to the readers.
>
> ___
>
>
> We hope our explanations have addressed your concerns. As we are in the discussion phase, we welcome any additional comments or questions regarding our response or the main paper. If further clarification is needed, please do not hesitate to mention it, and we will promptly address your inquiries. We look forward to receiving your feedback.
>
> Best regards,
>
> The Authors

---

### Official Review · Reviewer_xbfe · 2024-07-14

**Soundness:** 3
**Presentation:** 3
**Contribution:** 3
**Rating:** 7
**Confidence:** 4

**Summary:**

The paper introduces BIR-D, a novel approach utilizing generative diffusion models for blind image restoration without requiring predefined degradation types. Traditional methods assume degradation models and optimize their parameters, limiting their applicability. BIR-D overcomes this by employing an optimizable convolutional kernel that simulates degradation dynamically during diffusion steps, allowing it to handle various complex degradations.

**Strengths:**

The method stands out by integrating an optimizable convolutional kernel to dynamically adapt the degradation model during the diffusion steps, a concept not previously explored in the literature.

The introduction of an empirical formula for adaptive guidance scale is innovative, eliminating the need for manual grid searches and enhancing the practicality of the approach across diverse image restoration tasks.

The experimental results are robust, covering both qualitative and quantitative analyses on real-world and synthetic datasets. The superiority of BIR-D over existing methods is clearly demonstrated through comprehensive experimentation.

**Weaknesses:**

The reviewer appreciates the innovative use of an optimizable convolutional kernel to dynamically adapt the degradation model during the diffusion steps. This is considered the most significant contribution of the work. However, this section lacks sufficient analysis and visualization. While the paper asserts that GDP [1] assumes specific degradation types and is not suitable for complex degradation models, the differences and improvements of the proposed degradation model compared to the one in GDP are not clearly explained.

Additionally, the paper is missing some relevant references for blind IR [2] and earlier generative prior-based IR methods [3,4]. Overall, the reviewer appreciates the work and would be happy to adjust the rating if the aforementioned concerns are addressed.

[1] Generative diffusion prior for unified image restoration and enhancement. CVPR'23

[2] AND: Adversarial neural degradation for learning blind image super-resolution. NeurIPS'23

[3] Image restoration with deep generative models. ICASSP'18

[4] Maximum a posteriori on a submanifold: a general image restoration method with gan. IJCNN'20

**Questions:**

See weaknesses.

**Limitations:**

The authors have addressed the limitations.

---

> ### Author Rebuttal · Authors · 2024-08-06
>
> We sincerely thank Reviewer xbfe for devoting time to this review and providing valuable comments.
>
> > **Q1:**  (i)"The reviewer appreciates the innovative use of an optimizable convolutional kernel to dynamically adapt the degradation model during the time steps. This is considered the most significant contribution of the work. However, this section lacks sufficient analysis and visualization."
> (ii)"The differences and improvements of the proposed degradation model compared to the one in GDP are not clearly explained."
>
> **A:** We thank the reviewer for the comment.
>
> For question(i), in order to visualize the variation trends of convolution kernel parameters and guidance scale in the reverse process, we conducted experiments on the test set of the LOL dataset from the low-light enhancement task. As shown in Global PDF Figures 1 and 2, the mean values of the convolution kernel parameters are given by random initial values and gradually increase with the progress of the time steps. This trend is also consistent with actual degradation, causing the convolution kernel parameters to gradually approximate actual degradation. The gradient of the distance metric with respect to the convolution kernel parameters ensures our BIR-D in updating the convolution kernel parameters. Global PDF Figure 4 and 5 displays the changing of the degradation mask during the sampling process. This degradation mask learns detailed information from BIR-D. Additionally, incorporating the degradation mask helps BIR-D restore areas with significant brightness differences.
>
> Meanwhile, during the sampling process, the guidance scale gradually decreases as shown in Global PDF Figure 3. Therefore, at the end of the sampling process, the degree of adding guidance should be relatively low. The reason is that the change in $x_t$ in each step $t$ at the end of the sampling process is relatively small, resulting in the gradient term in the empirical formula also decreasing. This is also consistent with theoretical analysis and experimental results.
>
> For question (ii), Our main improvements compared to GDP in this article are as follows.
> 1. Different settings of degradation function
>     * The BIR-D proposed in our paper does not require an assumed degradation type or a given initial value for the degradation function. The degradation function is obtained by real-time updating of the optimizable parameters in the convolution kernel and mask during the time step.
>     But in some experimental tasks in GDP such as deblurring, super-resolution, inpainting, and colorization, the types and parameters of the degradation function need to be specified and remain unchanged in the time step. This also means that GDP is not suitable for simulating the degradation function through sampling for complex degradation models.
>     * In order to achieve blind image restoration performance, GDP assumes that the degradation form is $Y=fX+\mathcal{M}$. But this assumption is only valid for low light enhancement and HDR image restoration tasks, as it is only multiplied by a coefficient $f$ for all pixels in the image. BIR-D is more flexible in using convolutional kernels and masks, which can simulate more unknown degradation.
>     * GDP can only perform a combination of two types of degradation due to its degradation setting. Meanwhile, as shown in the teaser of GDP, it cannot handle the combination of super-resolution and deblur that simultaneously affects image quality. The degradation function of BIR-D is simulated using a convolutional kernel with optimizable parameters, which can flexibly and effectively solve 3-4 mixed degradation problems and the combination of super-resolution and deblur that simultaneously affects image quality.
>
> 2. The way of setting the guidance scale is different.
>     * In GDP, the guidance scale can only be set as a hyperparameter and remain unchanged during the sampling process. However, for different images in different tasks even the single image in different reverse steps, the theoretical values of the guidance scale should be different. The deviation of guidance scale values will greatly affect the quality of image restoration. For example, larger values will lead to the appearance of mineral textures in the results, as shown in global PDF Figure 6.
>     * In our paper, we propose an empirical formula for the adaptive guidance scale, which can be updated in real-time at each time step based on specific images in the BIR-D. This improvement avoids the complexity and bias of human settings, while also enhancing the model's restoration performance, which is also validated in the ablation study in the main paper.
> ___
>
> > **Q2:** "The paper is missing some relevant references for blind IR and earlier generative prior-based IR methods."
>
> **A:** Thanks for your suggestion. We have carefully read the four papers you provided. The proposed image restoration methods demonstrate a high level of creativity and value, significantly advancing the field of blind image restoration. We all agree that adding these four articles as references would be definitely helpful for the article, and we will incorporate them into the article in future versions.

---

> > ### Comment · Reviewer_xbfe · 2024-08-11
> >
> > Thank you for providing the additional details and clarifications in your response. I believes that the visualizations provided in the rebuttal file could help potential readers better understand the paper's contributions. The comparison to the GDP method also addresses my concerns. Therefore, I have increased my rating by one point.

---

> ### Author Response · Authors · 2024-08-11
> **Official Comment by Authors**
>
> We sincerely appreciate your thought-provoking reviews and are pleased to see your upgrading decision. Following your valuable suggestions, we will carefully incorporate these revisions into our future version. To substantiate our results, we will release the code. Thank you once again for your positive rating and the time devoted to this review.
>
> Best regards,
>
> The Authors

---

### Author Rebuttal · Authors · 2024-08-06

We are very grateful to all the reviewers for their valuable comments and suggestions on this article.

___

We are glad to see the reviewers' recognition of our work.
* "The method stands out by integrating an optimizable convolutional kernel to dynamically adapt the degradation model during the time steps, a concept not previously explored in the literature."(Reviewer xbfe)
* "The paper presents a robust validation of the BIR-D method through a comprehensive set of experiments across multiple image restoration tasks." (Reviewer CUUT)
* "The introduction of an empirical formula for adaptive guidance scale is innovative." (Reviewer xbfe)
* "The way to control the guidance scale is fancy." (Reviewer rhDr)

___

We would like to emphasize once again the innovation and main contribution of the article.
* We propose a universal blind image restoration model BIR-D, which utilizes an optimizable convolutional kernel to simulate the degradation model and dynamically update the parameters of the degradation model during the sampling process.
* We have provided an empirical formula for the chosen of adaptive guidance scale, eliminating the need for a grid search for the optimizable parameter compared with existing guided diffusion methods.
* BIR-D has demonstrated superior practicality and generality in various blind image restoration tasks in the real world and synthetic datasets compared to off-the-shelf unsupervised methods, both qualitatively and quantitatively.

___

We have made the following modifications and explanations to the manuscripts based on the suggestions and comments of the reviewers.
* We have supplemented the trends of parameters in the adaptive guidance scale and optimizable convolution kernel in the sampling process to better clarify how these designs contribute to the BIR-D's performance.
* We have listed and analyzed the advantages and improvement of BIR-D compared to GDP from multiple perspectives. Importantly, we re-clarify that the several challenges that remain in GDP are well solved by our BIR-D.
* We have explicated the necessity of proposing empirical formulas with guidance scales and optimized the derivation approach and process.
* We have provided more details on multi-guidance blind image restoration, including diagrams and pseudocode.
* We have polished the main text and clarified some typos and misunderstandings in the main submitted materials.

___

Last but not least, thanks again to PCs, ACs, and all reviewers for their time and effort in reviewing.

---

### Decision · Program_Chairs · 2024-09-25

**Decision:**

Accept (poster)

**Comment:**

This paper received an Accept, a Borderline Accept and a Reject. The reviewers appreciate the technical contributions of the paper and its novelty. The main concerns raised by the reviewers are: 1) Missing references, 2) Missing analysis and presentation of the method, 3) Technical issues with the method and experiments. The authors submitted a rebuttal and engaged in a discussion with the reviewers. The AC has read the rebuttal, the reviews, and the discussions. Two reviewers were satisfied with the rebuttal and discussion and raised their rating. The AC finds that the authors adequately clarified all the concerns raised by the reviewers and recommends the paper for acceptance.